# Release of cholesterol-rich particles from the macrophage plasma membrane during movement of filopodia and lamellipodia

Xuchen Hu[1], Thomas A Weston[1], Cuiwen He[1], Rachel S Jung[1], Patrick J Heizer[1], Brian D Young[2], Yiping Tu[1], Peter Tontonoz[3], James A Wohlschlegel[2], Haibo Jiang[1,4]*, Stephen G Young[1,5]*, Loren G Fong[1]*

[1]Department of Medicine, David Geffen School of Medicine, University of California, Los Angeles, Los Angeles, United States; [2]Department of Biological Chemistry, University of California, Los Angeles, Los Angeles, United States; [3]Department of Pathology and Laboratory Medicine, University of California, Los Angeles, Los Angeles, United States; [4]School of Molecular Sciences, University of Western Australia, Perth, Australia; [5]Department of Human Genetics, David Geffen School of Medicine, University of California, Los Angeles, Los Angeles, United States

**Abstract** Cultured mouse peritoneal macrophages release large numbers of ~30-nm cholesterol-rich particles. Here, we show that those particles represent fragments of the plasma membrane that are pulled away and left behind during the projection and retraction of filopodia and lamellipodia. Consistent with this finding, the particles are enriched in proteins found in focal adhesions, which attach macrophages to the substrate. The release of particles is abolished by blocking cell movement (either by depolymerizing actin with latrunculin A or by inhibiting myosin II with blebbistatin). Confocal microscopy and NanoSIMS imaging studies revealed that the plasma membrane–derived particles are enriched in 'accessible cholesterol' (a mobile pool of cholesterol detectable with the modified cytolysin ALO-D4) but not in sphingolipid-sequestered cholesterol [a pool detectable with ostreolysin A (OlyA)]. The discovery that macrophages release cholesterol-rich particles during cellular locomotion is likely relevant to cholesterol efflux and could contribute to extracellular cholesterol deposition in atherosclerotic plaques.
DOI: https://doi.org/10.7554/eLife.50231.001

*For correspondence:
haibo.jiang@uwa.edu.au (HJ);
sgyoung@mednet.ucla.edu (SGY);
lfong@mednet.ucla.edu (LGF)

## Introduction

A key function of macrophages is to engulf and digest cellular debris. The cholesterol in the debris can be esterified and stored in cytosolic lipid droplets (*Brown et al., 1980*), thereby avoiding toxicity associated with an accumulation of free cholesterol, but macrophages ultimately must dispose of the surplus cholesterol, a process generally referred to as 'cholesterol efflux' (*Rosenson et al., 2012*; *Tall et al., 2002*; *Rothblat and Phillips, 2010*; *Adorni et al., 2007*). One mechanism for cholesterol efflux involves transferring free cholesterol to high density lipoproteins (HDL), a process that is facilitated by ABC transporters (*Rosenson et al., 2012*; *Tall et al., 2002*; *Rothblat and Phillips, 2010*; *Yvan-Charvet et al., 2010*; *Westerterp et al., 2014*; *Duong et al., 2006*). A deficiency of ABCA1 interferes with cholesterol efflux by macrophages, leading to an accumulation of foam cells (macrophages containing numerous cholesterol ester droplets) in tissues (*Brooks-Wilson et al., 1999*; *Bodzioch et al., 1999*; *Clee et al., 2001*; *Rust et al., 1999*). Another potential mechanism for cholesterol efflux by macrophages is the release of cholesterol-rich particles from the plasma membrane. Using a cholesterol-specific monoclonal antibody and immunocytochemical approaches, the laboratory of Howard Kruth reported that cultured human monocyte–derived macrophages release

'cholesterol microdomains' (varying in size but as large as several hundred nm) onto the surrounding substrate (*Jin et al., 2016*; *Freeman et al., 2014*; *Ong et al., 2010*; *Jin et al., 2015*; *Jin et al., 2018*). The release of these microdomains was reduced by decreasing the expression of ABC transporters (*Jin et al., 2016*; *Freeman et al., 2014*; *Jin et al., 2015*; *Jin et al., 2018*). Recently, we demonstrated, by scanning electron microscopy (SEM), that large numbers of ~30-nm vesicular particles were released from the plasma membrane of mouse peritoneal macrophages and a mouse macrophage cell line by a process that morphologically resembles 'budding.' The plasma membrane–derived particles were enriched in 'accessible cholesterol' (*He et al., 2018*), a mobile pool of cholesterol that can be detected by ALO-D4, a modified cholesterol-binding cytolysin (*Gay et al., 2015*). The cholesterol content of the particles could be increased by loading macrophages with cholesterol or by treating the cells with a liver X receptor (LXR) agonist, and the cholesterol content of the particles could be depleted with HDL (*He et al., 2018*).

The SEM studies by He and coworkers (*He et al., 2018*) demonstrated that the particles released by macrophages are derived from the plasma membrane, but the mechanism was unclear. One possibility is that the budding of particles from the plasma membrane was driven by shuttling cholesterol into plasma membrane microdomains, causing outward ballooning of a localized segment of the plasma membrane and ultimately to the release of a vesicular particle. A second possibility, mentioned by He and coworkers (*He et al., 2018*) but not pursued, was that the particles represented segments of the macrophage plasma membrane that had been affixed to the substrate but then 'torn away and left behind' during movement of filopodia and lamellipodia.

In the current study, we used live-cell microscopy and SEM, along with proteomic studies, to explore the mechanism for the release of particles from the plasma membrane of macrophages. We also used super-resolution fluorescence microscopy and NanoSIMS imaging, in combination with two different cholesterol-binding proteins (one specific for 'accessible cholesterol' and the other for sphingomyelin-sequestered cholesterol), to examine the cholesterol pools within the plasma membrane–derived particles released by macrophages.

## Results

### Macrophages release plasma membrane–derived particles during the movement of filopodia and lamellipodia

In the current studies, we again found that particles are released from macrophage filopodia and lamellipodia onto the surrounding substrate by a process that resembles budding (*Figure 1—figure supplement 1*). Because the released particles remain adherent to the substrate, we suspected that the particles might represent fragments of the plasma membrane that were affixed to the underlying substrate but then pulled away and left behind during movement of filopodia/lamellipodia. To explore this idea, we plated mouse peritoneal macrophages onto gridded glass-bottom dishes and recorded images of cells by live-cell microscopy (*Figure 1—videos 1–2*), making it possible to visualize the projection and retraction of filopodia/lamellipodia. The same cells were then fixed and imaged by SEM. Lawns of ~30-nm particles were present on the substrate surrounding macrophages, generally at the lagging pole of the cell and invariably where we had observed, by live-cell imaging, the extension and retraction of filopodia/lamellipodia (*Figure 1*, *Figure 1—videos 1–2*).

To determine whether the extension and retraction of filopodia/lamellipodia are required for particle release, cell movement was blocked by treating macrophages with an actin-depolymerizing agent (latrunculin A) or a myosin II inhibitor (blebbistatin). Live-cell imaging revealed that the drug-treated macrophages were unable to project and retract filopodia/lamellipodia (*Figure 2—videos 1–6*). Macrophages were incubated with latrunculin A or blebbistatin in suspension for 1 hr ('pre-treatment') and then plated onto poly-D-lysine–coated silicon wafers and incubated with the drugs overnight (*Figure 2*). Both latrunculin A and blebbistatin abolished particle release from 'pre-treatment' macrophages (*Figure 2A–B*). Other macrophages were allowed to adhere to the substrate for 1 hr before initiating drug treatment ('post-adherence') (*Figure 2A–B*). In latrunculin A–treated 'post-adherence' macrophages, a circumferential ring of particles was left behind on the substrate as the cell retracted with actin depolymerization (*Figure 2A*). Blebbistatin eliminated particle release in post-adherence cells (*Figure 2B*). Macrophages treated with vehicle alone (DMSO) released large numbers of particles onto the surrounding substrate (*Figure 2A–B*). As an additional control,

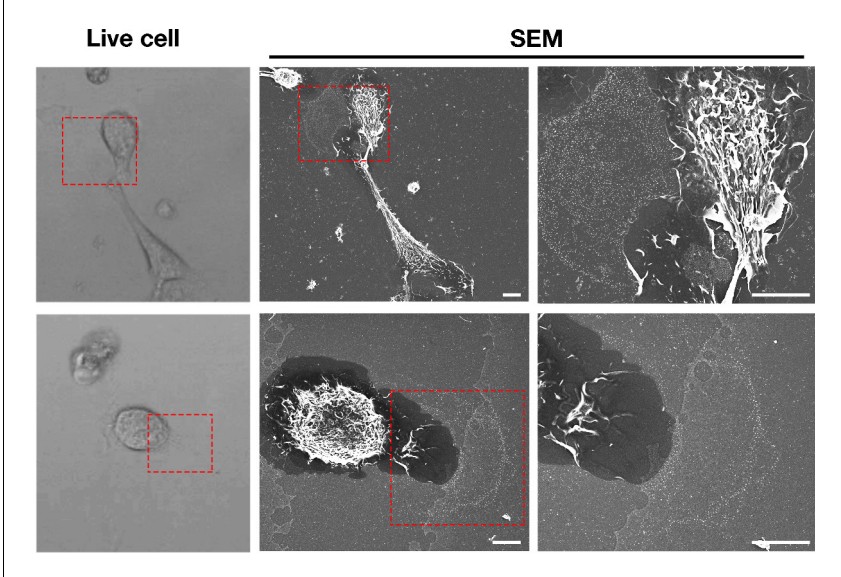

**Figure 1.** Macrophages release plasma membrane–derived particles onto the substrate during extension and retraction of filopodia and lamellipodia, as judged by correlative live-cell imaging and SEM. Cells were plated onto poly-D-lysine–coated gridded glass-bottom Petri dishes, and videos were recorded for 24 hr at 5 min intervals (see *Figure 1—videos 1–2*). The 'Live cell' images show the final frame of the videos. The imaging of cells by SEM made it possible to visualize a lawn of particles that had been released onto the substrate during the projection and retraction of filopodia/lamellipodia. The *red* boxed region in the live-cell image and in the low-magnification SEM image is shown in the SEM image on the far right. Three independent experiments were performed; representative images are shown. Scale bar, 5 μm.

DOI: https://doi.org/10.7554/eLife.50231.002

The following video and figure supplement are available for figure 1:

**Figure supplement 1.** Macrophages release particles from the plasma membrane of filopodia and lamellipodia by a process that resembles budding.

DOI: https://doi.org/10.7554/eLife.50231.003

**Figure 1—video 1.** Mouse peritoneal macrophages release vesicular particles onto the surrounding substrate during the extension and retraction of filopodia and lamellipodia.

DOI: https://doi.org/10.7554/eLife.50231.004

**Figure 1—video 2.** Shows a macrophage imaged by SEM in the bottom row of *Figure 1*.

DOI: https://doi.org/10.7554/eLife.50231.005

macrophages that had been incubated with drugs overnight were washed and incubated for an additional 18 hr without drugs. In the absence of the drugs, the morphology of the cells returned to normal and particle release resumed, with numerous particles appearing on the surrounding substrate (*Figure 2A–B*).

To determine if the particles that are released during the projection and retraction of lamellipodia contain 'accessible cholesterol' (a mobile pool of cholesterol that is *not* sequestered by sphingolipids), we performed live-cell imaging of RAW 264.7 macrophages (*Figure 3—videos 1–2*) and then incubated the cells with [$^{15}$N]ALO-D4. The macrophages were then processed for SEM and Nano-SIMS imaging. The lawn of particles around macrophages, visible by SEM, was enriched in $^{15}$N, as revealed by NanoSIMS imaging (*Figure 3*). Of note, the degree of $^{15}$N enrichment was greater in the lawn of particles than on the plasma membrane covering the macrophage cell body or macrophage filopodia (*Figure 3*).

## Macrophage particles contain plasma membrane proteins

Given that the particles on the substrate were derived from the plasma membrane, we suspected that they would contain plasma membrane proteins. Two findings lent credence to our suspicion. First, after biotinylating cell-surface proteins of macrophages with Sulfo-NHS-SS-biotin, the lawn of

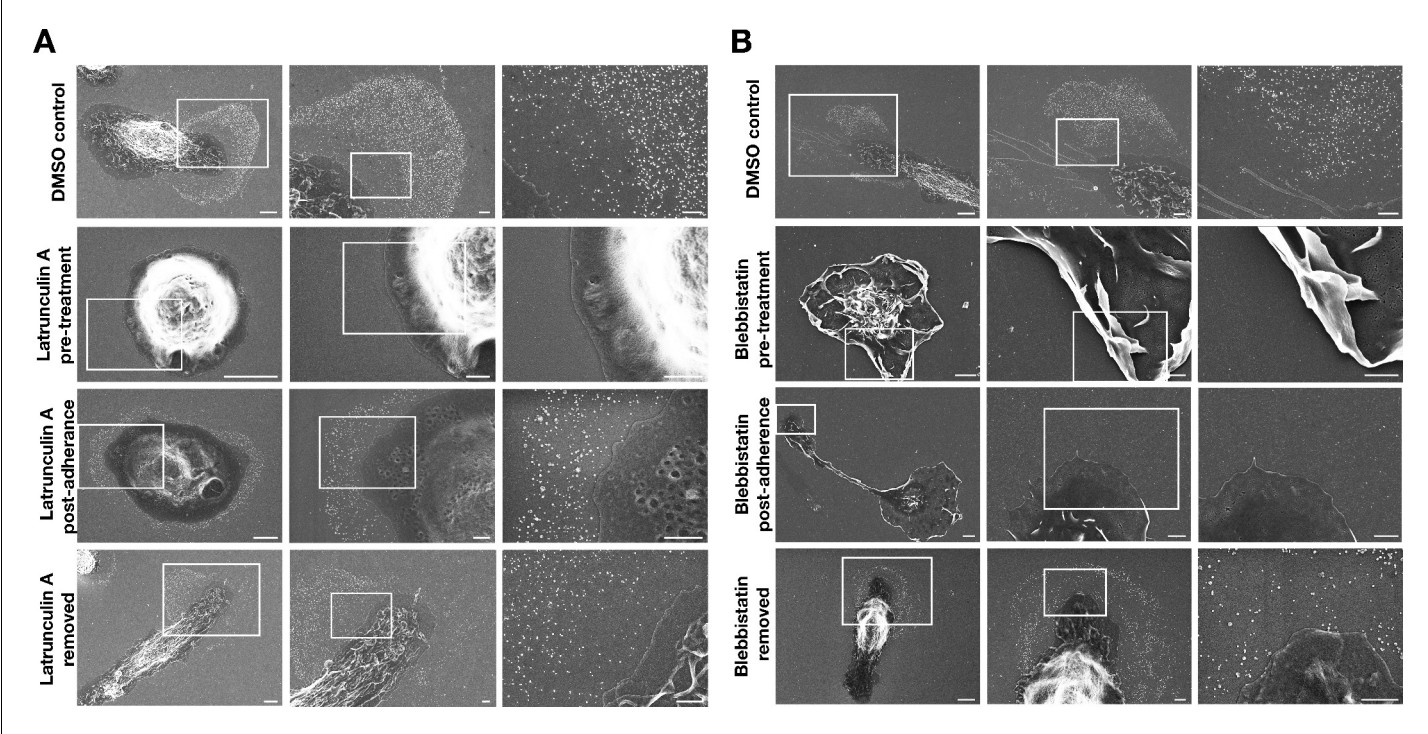

**Figure 2.** Inhibiting macrophage movement with latrunculin A or blebbistatin abolishes particle release onto the surrounding substrate. Scanning electron micrographs (SEMs) of mouse peritoneal macrophages that had been treated with latrunculin A (**A**) or blebbistatin (**B**). Macrophages were treated with latrunculin A, blebbistatin, or vehicle alone (DMSO) overnight; the drug treatments were initiated when the cells were in suspension 1 h before plating ('pre-treatment') or 1 h after adherence to the substrate ('post-adherence'). Macrophages treated with vehicle alone (DMSO) released particles onto the substrate. No particles were visible on the substrate in cells that had been pre-treated with latrunculin A or blebbistatin—or in the blebbistatin post-adherence cells. In the latrunculin A post-adherence cells, a circumferential ring of particles was observed around cells. Latrunculin A and blebbistatin were removed from some dishes after the overnight incubation, and the cells were incubated for an additional 18 h without drugs. After removing the drugs, the release of particles onto the substrate resumed. Four independent experiments were performed; representative images are shown. Scale bars for images on the left in each panel are 4 μm; scale bars for the images in the middle and right are 1 μm.

DOI: https://doi.org/10.7554/eLife.50231.006

The following videos are available for figure 2:

**Figure 2—video 1.** Treatment of mouse peritoneal macrophages with latrunculin A or blebbistatin prevent projection and retraction of filopodia/lamellipodia.

DOI: https://doi.org/10.7554/eLife.50231.007

**Figure 2—video 2.** Shows macrophages treated with vehicle (DMSO) alone.

DOI: https://doi.org/10.7554/eLife.50231.008

**Figure 2—video 3.** Shows macrophages treated with latrunculin A.

DOI: https://doi.org/10.7554/eLife.50231.009

**Figure 2—video 4.** Shows macrophages treated with latrunculin A.

DOI: https://doi.org/10.7554/eLife.50231.010

**Figure 2—video 5.** Shows a macrophage treated with blebbistatin.

DOI: https://doi.org/10.7554/eLife.50231.011

**Figure 2—video 6.** Shows macrophages treated with blebbistatin.

DOI: https://doi.org/10.7554/eLife.50231.012

particles surrounding macrophages could be detected with fluorescent streptavidin (colocalizing with fluorescently labeled ALO-D4, which binds 'accessible cholesterol') (*Figure 4A*). Also, by SEM, streptavidin-conjugated 40-nm gold nanoparticles bound to both macrophages and to the surrounding particles (*Figure 4B*). There was no binding of the gold nanoparticles to non-biotinylated macrophages (*Figure 4—figure supplement 1*). Second, by NanoSIMS analyses, the lawn of particles outside macrophages contained $^{14}$N and $^{32}$S (as well as accessible cholesterol, detectable with [$^{15}$N] ALO-D4) (*Figure 4—figure supplement 2*). In light of these findings, we prepared both particle and

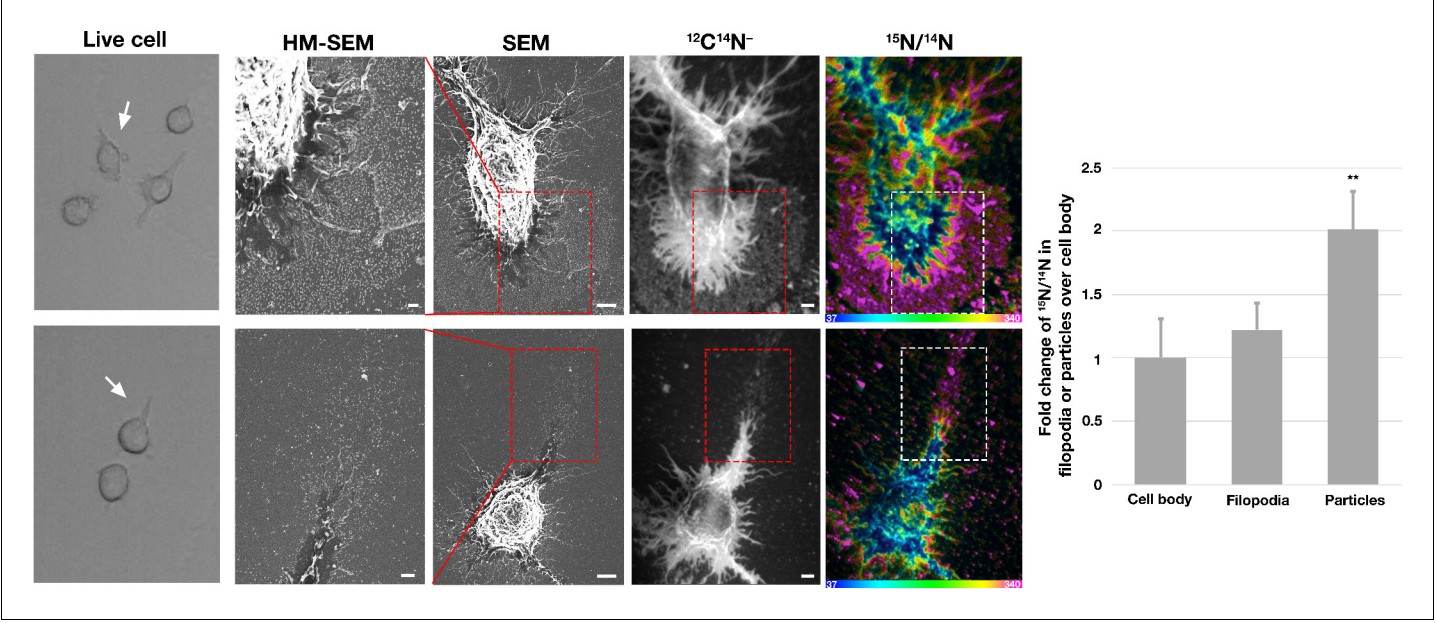

**Figure 3.** Correlative live-cell, scanning electron microscopy (SEM), and NanoSIMS imaging, revealing that particles released onto the substrate during movement of filopodia and lamellipodia are enriched in accessible cholesterol. RAW 264.7 macrophages were plated onto iridium- and poly-D-lysine–coated gridded glass-bottom Petri dishes. Videos were recorded for 24 hr at 5 min intervals (see *Figure 3—videos 1–2*). The 'Live cell' images in this figure show the final frame of the videos, with the *white* arrows pointing to the cells that were subsequently visualized by SEM and NanoSIMS. After live-cell imaging, cells were incubated with [$^{15}$N]ALO-D4 (a modified cytolysin that binds to 'accessible cholesterol'). The same cells that were imaged by live-cell imaging were subsequently imaged by SEM (to visualize particles) and NanoSIMS (to visualize [$^{15}$N]ALO-D4 binding). The particles left behind on the substrate during movement of lamellipodia and filopodia bound [$^{15}$N]ALO-D4 avidly. $^{12}$C$^{14}$N$^-$ NanoSIMS images were used to visualize cell morphology; the $^{15}$N/$^{14}$N images show $^{15}$N enrichment (*i.e.*, binding of [$^{15}$N]ALO-D4). The boxed region in the SEM and NanoSIMS images is shown at higher magnification in the 'HM-SEM' image, providing definition of individual particles. $^{15}$N enrichment was ~twofold higher in the macrophage particles than on the plasma membrane over the cell body and the filopodia. Two independent experiments were performed; representative images are shown. Quantification of $^{15}$N/$^{14}$N ratios were performed on the cell body and macrophage particles [25 distinct regions of the cell body, filopodia, and particles (20 pixels in diameter) were circled on the $^{12}$C$^{14}$N$^-$ image, followed by calculation of $^{15}$N/$^{14}$N ratios in each region]. Images from six macrophages were used for the quantification. Graph shows the mean and standard deviation of the fold change of $^{15}$N enrichment in particles and filopodia, normalized to the macrophage cell body. **$p < 0.001$. Scale bar, 2 μm.

DOI: https://doi.org/10.7554/eLife.50231.013

The following videos are available for figure 3:

**Figure 3—video 1.** RAW 264.7 macrophages release particles during projection and retraction of lamellipodia.

DOI: https://doi.org/10.7554/eLife.50231.014

**Figure 3—video 2.** Shows a macrophage imaged by SEM and NanoSIMS in the bottom row of *Figure 3*.

DOI: https://doi.org/10.7554/eLife.50231.015

plasma membrane preparations from RAW 264.7 mouse macrophages (see Materials and methods) for shotgun proteomics. By negative-stain transmission electron microscopy (TEM), the size of particles in the particle preparation (*Figure 5—figure supplement 1B*) was similar to particles in the SEM images (*Figure 5—figure supplement 1A*). TEM images of the plasma membrane preparations revealed aggregates of membranous material (*Figure 5—figure supplement 1B*). Proteomic studies on three independent particle and plasma membrane preparations revealed that both were enriched in focal adhesion proteins and cytoskeletal components (*Figure 5A–B*). When analyses were confined to the top 75th percentile of proteins by spectral count, we identified 653 proteins in the particle preparations and 715 proteins in the plasma membrane preparations, with 502 proteins in common (*Figure 5C*). The top 15 focal adhesion–related proteins, as annotated by Gene Ontology, were found in both the particle and the plasma membrane preparations, but the majority of those proteins were relatively more abundant in the particle preparations (*Figure 5D*).

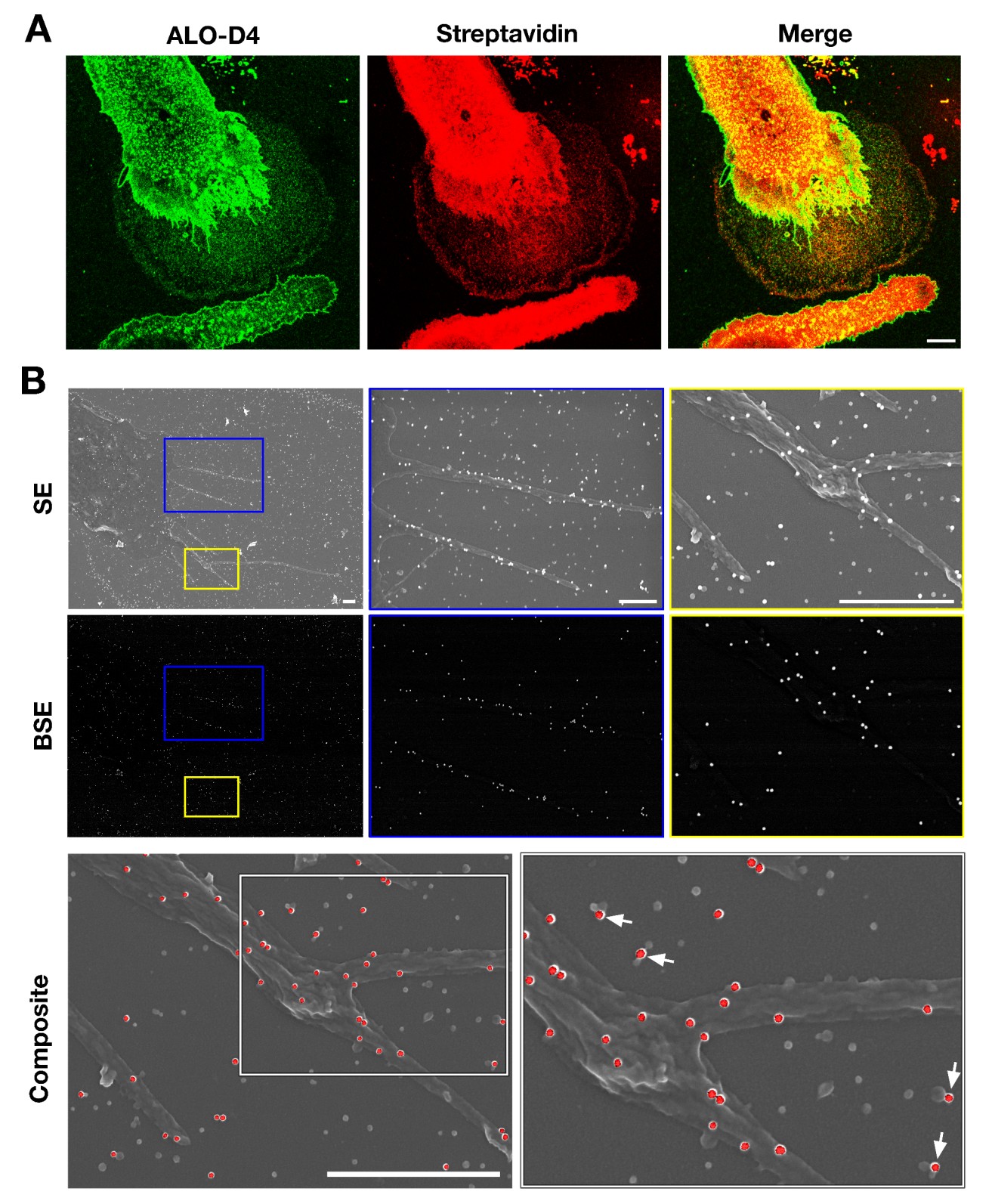

**Figure 4.** Particles released from the plasma membrane of biotinylated mouse peritoneal macrophages can be detected with streptavidin and ALO-D4 (a cytolysin that binds to the accessible pool of cholesterol). (A) Biotinylated macrophage particles can be detected with fluorescently labeled streptavidin as judged by confocal microscopy. Macrophages in suspension were biotinylated with Sulfo-NHS-SS-biotin. After plating the cells onto glass coverslips and incubating the cells in macrophage growth medium containing 10% FBS, the cells were washed and then incubated with Atto

*Figure 4 continued on next page*

*Figure 4 continued*
647N–labeled streptavidin (*red*) and Alexa Fluor 488–labeled [$^{15}$N]ALO-D4 (*green*). Cells were then fixed with 3% PFA and imaged by super-resolution stimulated emission depletion (STED) microscopy. Streptavidin and ALO-D4 bound to the macrophages as well as the lawn of particles on the surrounding substrate. Scale bar, 5 µm. (B) Biotinylated macrophage particles can be detected with streptavidin-conjugated gold nanoparticles, as judged by scanning electron microscopy (SEM). After biotinylating the plasma membrane of mouse peritoneal macrophages with Sulfo-NHS-SS-biotin, cells were plated onto glass-bottom Petri dishes. On the following day, the cells were incubated with streptavidin-conjugated 40-nm gold nanoparticles. Cells were then fixed with 1% glutaraldehyde and processed for SEM. Secondary electron (SE) and backscattered electron (BSE) images revealed gold nanoparticles on the macrophage cell body, filopodia, and on macrophage particles that had been released onto the substrate. Higher magnification images of the *blue* and *yellow* boxes are shown on the right. Composite images (*gray*, SE; *red*, BSE) images show colocalization of gold nanoparticles with the particles on the substrate. Higher magnification image of the *white* box is shown on the right. *White* arrows point to gold nanoparticles binding to macrophage particles. Three independent experiments were performed; representative images are shown. Scale bar, 1 µm.
DOI: https://doi.org/10.7554/eLife.50231.016
The following figure supplements are available for figure 4:

**Figure supplement 1.** Particles released from non-biotinylated macrophages do not bind streptavidin-conjugated gold nanoparticles as judged by scanning electron microscopy (SEM).
DOI: https://doi.org/10.7554/eLife.50231.017
**Figure supplement 2.** Correlative SEM and NanoSIMS imaging of macrophages and plasma membrane–derived particles on the surrounding substrate.
DOI: https://doi.org/10.7554/eLife.50231.018

## Inhibition of focal adhesion disassembly increases release of particles

Focal adhesions are macromolecular assemblies that link the actin cytoskeleton inside cells to the extracellular substrate (*Lauffenburger and Horwitz, 1996*; *Burridge and Fath, 1989*). The presence of focal adhesion proteins and cytoskeletal proteins in the particles is consistent with the observation that particles are released when small segments of the plasma membrane are pulled away and left behind during movement of filopodia/lamellipodia. We suspected that particle release might be increased by interfering with disassembly of focal adhesions. Focal adhesion kinase (FAK) is important for focal adhesion disassembly. Phosphorylation of Tyr-397 in FAK is one of the events in initiating focal adhesion disassembly (*Hamadi et al., 2005*; *Nagano et al., 2012*), and that step can be blocked with an FAK inhibitor. In macrophages treated with an FAK inhibitor, the filopodia and lamellipodia remain mobile (*Figure 6—videos 1–2*). In four different experiments, the lawn of particles surrounding macrophages appeared larger in cells treated with an FAK inhibitor (*Figure 6— figure supplement 1*). Indeed, the lawns of particles outside of FAK inhibitor–treated macrophages were ~twice the size of the macrophage cell body, whereas they were only ~40% as large as the cell body in DMSO-treated macrophages (10 macrophages were analyzed per group). Cholesterol-loaded macrophages (loaded with an incubation with acetylated low-density lipoproteins) retained their mobility, as judged by live-cell imaging (*Figure 6—videos 3–4*), and lawns of particles were present on the substrate surrounding cholesterol-loaded cells (*Figure 6—figure supplement 1*). When cholesterol-loaded macrophages were treated with an FAK inhibitor, large lawns of particles were observed around almost every cell (*Figure 6—figure supplement 1*).

## Macrophage-derived particles are enriched in accessible cholesterol but not sphingolipid-sequestered cholesterol

Next, we asked whether the particles surrounding macrophages were enriched in sphingomyelin-sequestered cholesterol. Initially, we used super-resolution microscopy to compare the binding of ALO-D4 (which binds to accessible cholesterol) (*Gay et al., 2015*) and OlyA (which binds to sphingo-myelin-bound cholesterol) (*Endapally et al., 2019*) to macrophages and to particles on the surrounding substrate. Mouse peritoneal macrophages were plated onto glass-bottom MatTek dishes and incubated in medium containing an FAK inhibitor or vehicle alone (DMSO). After 24 hr, the cells were incubated at 4°C with Alexa Fluor 488–labeled [$^{15}$N]ALO-D4 and Atto 647N–labeled [$^{13}$C]OlyA. Stimulated emission depletion (STED) microscopy revealed avid ALO-D4, but not OlyA, binding to the lawn of particles around macrophages (*Figure 6*). The lawn of particles surrounding cells was larger in cells that had been treated with an FAK inhibitor, consistent with the SEM findings in *Figure 6—figure supplement 1*.

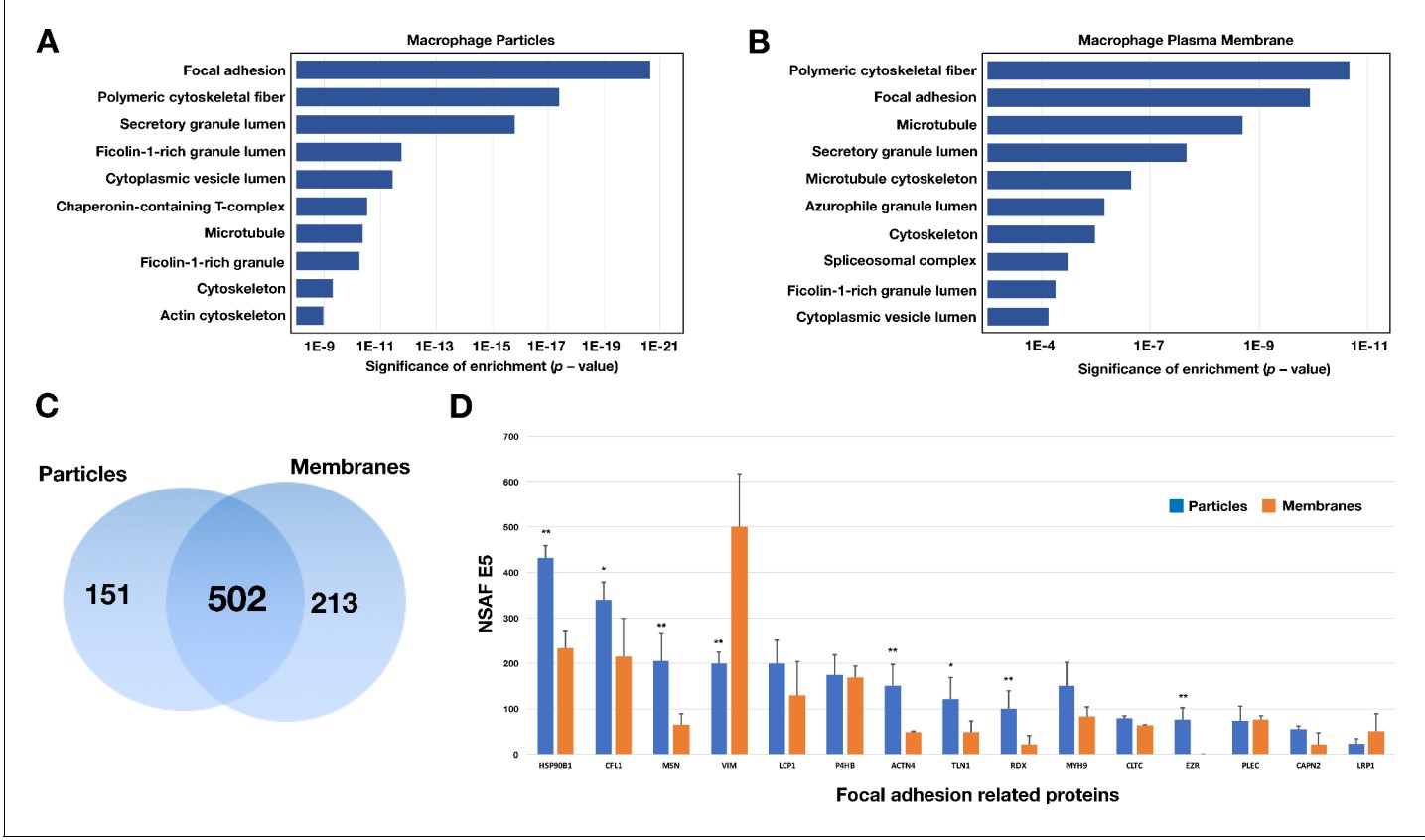

**Figure 5.** Enrichment in focal adhesion proteins in the particle preparations from RAW 264.7 macrophages. The most abundant proteins (the top 75th percentile by spectral counts) were analyzed by Enrichr and categorized by GO Cellular Components 2018. (A–B) Analysis of proteins in macrophage particles ($n$ = 653) and macrophage plasma membranes ($n$ = 715) by GO categories. The top 10 cellular component categories were ordered by level of statistical significance. (C) Venn diagram depicting numbers of proteins present in the particle preparation only, the plasma membrane preparation only, or both. (D) Bar graph showing the top 15 focal adhesion–related proteins by the normalized spectral abundance factor (NSAF), multiplied by 10,000. The particle fraction is shown in *blue*; the plasma membrane fraction is shown in *orange*. The bar graph shows the mean ± SD for three independent experiments. *$p$<0.05; **$p$<0.001.
DOI: https://doi.org/10.7554/eLife.50231.019

The following figure supplement is available for figure 5:

**Figure supplement 1.** Isolation of particles released onto the substrate by RAW 264.7 macrophages.
DOI: https://doi.org/10.7554/eLife.50231.020

In related experiments, we examined macrophages that had been incubated with latrunculin A for 1 hr in suspension and subsequently plated on poly-D-lysine–coated silicon wafers and incubated in drug-containing medium for 24 hr ('pre-treatment'). Because latrunculin A blocks particle release from macrophages, neither ALO-D4 nor OlyA detected particles on the substrate around macrophages (*Figure 6—figure supplement 2*). However, when latrunculin A was added to the medium after allowing the cells to adhere to the substrate ('post-adherence'), a circumferential ring of ALO-D4 binding, but not OlyA binding, was observed on the substrate surrounding macrophages (*Figure 6—figure supplement 2*), corresponding to the circumferential ring of particles detected by SEM in *Figure 2A*. In cells treated with DMSO alone, we observed ALO-D4, but not OlyA, binding to the lawn of particles around macrophages (*Figure 6—figure supplement 2*), similar to the results in *Figure 6*.

We suspected that the binding of lysenin (a cytolysin that binds sphingomyelin) would bind only weakly to the lawn of particles surrounding macrophages (resembling the findings with OlyA). Indeed, fluorescence microscopy of macrophages that had been incubated with Alexa Fluor 488–labeled [15N]ALO-D4 and mCherry-tagged lysenin (which binds sphingomyelin) revealed avid binding of ALO-D4, but not mCherry-tagged lysenin, to the lawn of particles surrounding macrophages

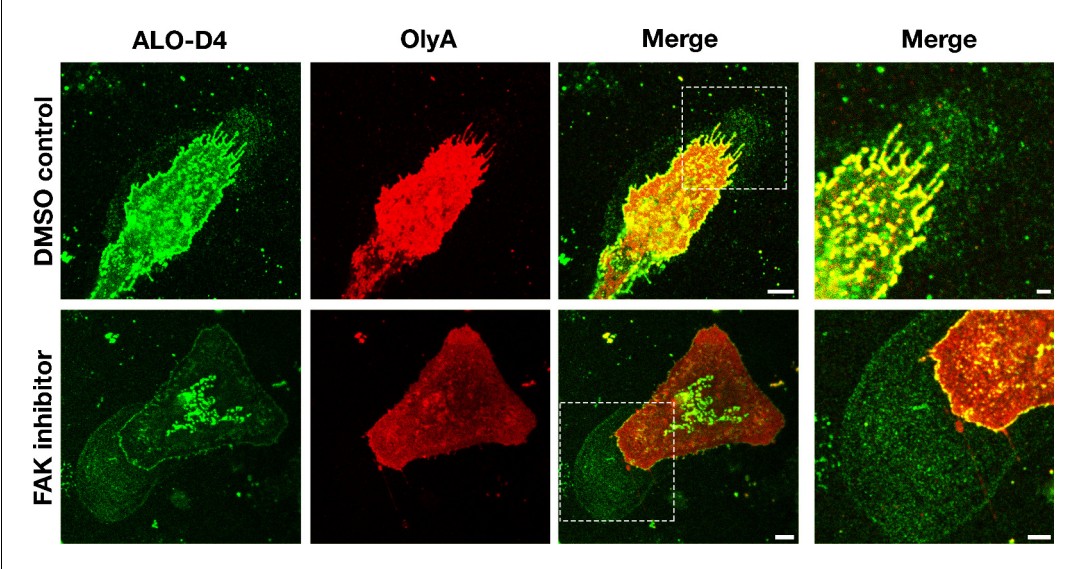

**Figure 6.** Particles released by mouse peritoneal macrophages onto the surrounding substrate are enriched in accessible cholesterol but not sphingomyelin-sequestered cholesterol. Mouse peritoneal macrophages were plated onto poly-D-lysine–coated glass coverslips and incubated overnight in medium containing 10% FBS and either an FAK inhibitor (CAS 4506-66-5, 2 μM) or vehicle (DMSO) alone. On the next day, the cells were incubated with Alexa Fluor 488–labeled [$^{15}$N]ALO-D4 (*green*), which binds to accessible cholesterol, and Atto 647N–labeled [$^{13}$C]OlyA (*red*), which binds to sphingomyelin-bound cholesterol (both at 20 μg/ml). Cells were then washed, fixed with 3% PFA, and imaged by STED microscopy. STED images were obtained from the bottom of the macrophage (optical section of ~200 nm). The lawn of particles surrounding macrophages was readily detectable with ALO-D4, but the binding of OlyA to particles was negligible. Two independent experiments were performed; representative images are shown. Scale bar, 5 μm. Higher magnification images of the boxed regions are shown on the right. Scale bar, 2 μm.

DOI: https://doi.org/10.7554/eLife.50231.021

The following video and figure supplements are available for figure 6:

**Figure supplement 1.** Inhibiting focal adhesion kinase in mouse peritoneal macrophages is accompanied by large lawns of particles on the surrounding substrate.

DOI: https://doi.org/10.7554/eLife.50231.022

**Figure supplement 2.** Incubating mouse peritoneal macrophages with latrunculin A alters the distribution of ALO-D4 binding.

DOI: https://doi.org/10.7554/eLife.50231.023

**Figure supplement 3.** Particles released by mouse peritoneal macrophages are enriched in accessible cholesterol but not in sphingomyelin.

DOI: https://doi.org/10.7554/eLife.50231.024

**Figure supplement 4.** Sphingomyelinase treatment reduces OlyA and lysenin binding to the plasma membrane.

DOI: https://doi.org/10.7554/eLife.50231.025

**Figure 6—video 1.** Movement of mouse peritoneal macrophages under different experimental conditions.

DOI: https://doi.org/10.7554/eLife.50231.026

**Figure 6—video 2.** Shows macrophages treated with an FAK inhibitor.

DOI: https://doi.org/10.7554/eLife.50231.027

**Figure 6—video 3.** Shows acLDL-loaded macrophages treated with vehicle (DMSO) alone.

DOI: https://doi.org/10.7554/eLife.50231.028

**Figure 6—video 4.** Shows acLDL-loaded macrophages treated with an FAK inhibitor.

DOI: https://doi.org/10.7554/eLife.50231.029

(*Figure 6—figure supplement 3*). When macrophages or CHO cells were treated with sphingomyelinase, the binding of OlyA and lysenin to cells was markedly reduced or abolished (*Figure 6—figure supplement 4*).

Next, we performed correlative live-cell, SEM, and NanoSIMS imaging, with the goal of comparing [$^{15}$N]ALO-D4 and [$^{13}$C]OlyA binding to the lawn of particles outside macrophages. The live-cell imaging allowed us to visualize the projection and retraction of lamellipodia (*Figure 7—videos 1–2*), and the SEMs revealed the expected lawn of particles outside cells (*Figure 7, Figure 7—figure*

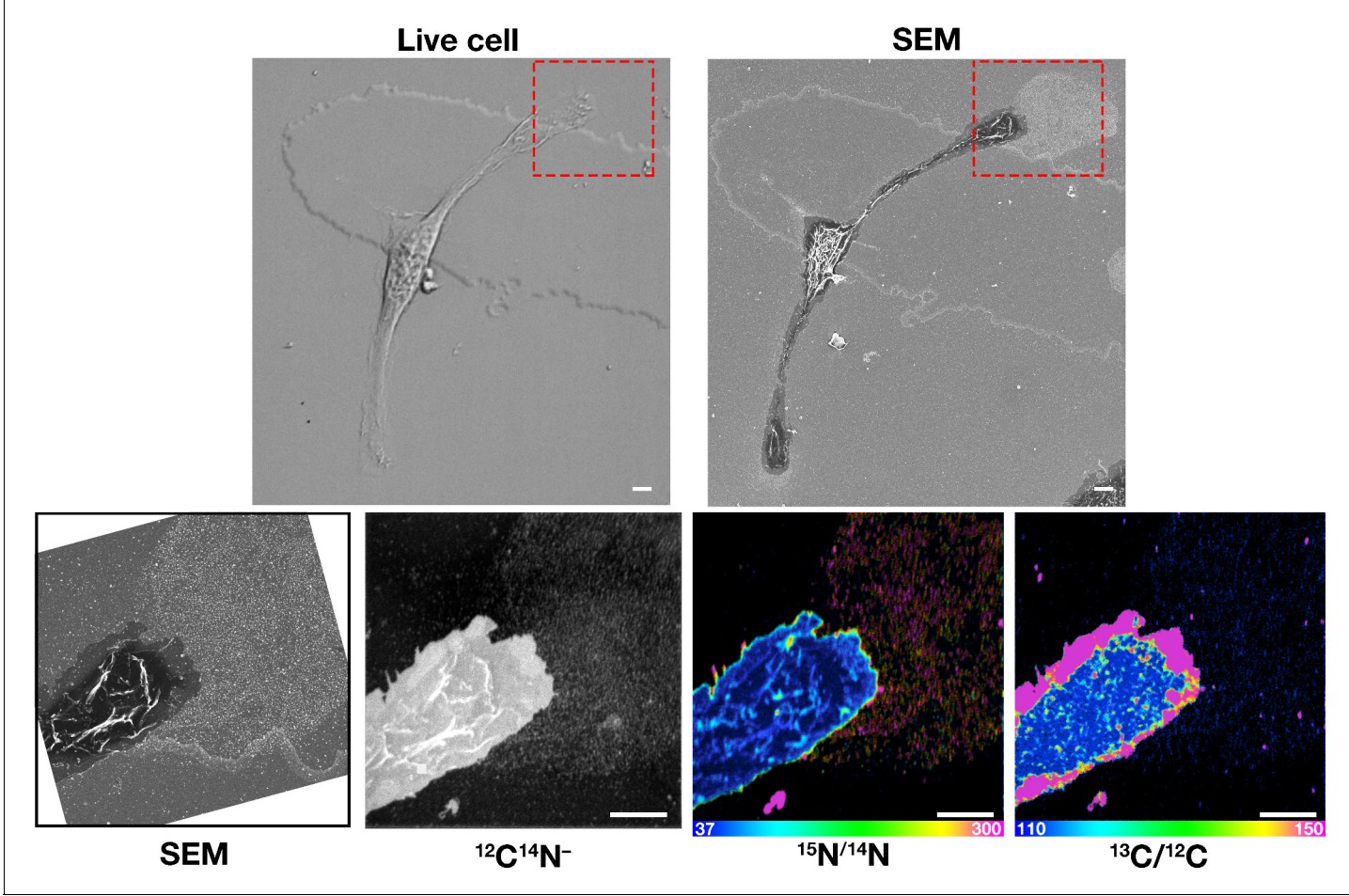

**Figure 7.** Correlative live-cell, SEM, and NanoSIMS imaging of mouse peritoneal macrophages, demonstrating that particles released onto the substrate during movement of filopodia and lamellipodia are enriched in accessible cholesterol but not sphingolipid-sequestered cholesterol. Macrophages were plated onto iridium- and poly-D-lysine–coated gridded glass-bottom Petri dishes, and videos of cell movement were recorded for 24 h at 5-min intervals (see *Figure 7—video 1*). The *red* arrow in the video points to an area of lamellipodia projection/retraction; the *white* box depicts the area that was later visualized by SEM and NanoSIMS. The 'Live cell' image in this figure shows the final frame of the video. After live-cell imaging, the cells were then incubated with [$^{15}$N]ALO-D4 (which binds to accessible cholesterol) and [$^{13}$C]OlyA (which binds to sphingomyelin-sequestered cholesterol). The cells were then imaged by SEM (to visualize particles) and then by NanoSIMS to visualize [$^{15}$N]ALO-D4 and [$^{13}$C]OlyA binding. The NanoSIMS images were created from secondary ions released from the top of the macrophage plasma membrane and the surrounding macrophage particles. Thus, the image differs from confocal fluorescence images in *Figure 6*, where we focused on the bottom surface of the macrophage (optical section of ~200 nm). The particles released onto the substrate were enriched in accessible cholesterol but not sphingomyelin-sequestered cholesterol. $^{12}$C$^{14}$N$^-$ images were useful for cell morphology; $^{15}$N/$^{14}$N images depict binding of [$^{15}$N]ALO-D4; $^{13}$C/$^{12}$C images depict binding of [$^{13}$C]OlyA. Here, the $^{13}$C/$^{12}$C and $^{15}$N/$^{14}$N scales were adjusted for visualization of macrophage-derived particles. Additional NanoSIMS images of this cell, with different scales, are shown in *Figure 8A*. Two independent experiments were performed; representative images are shown. Scale bar, 5 μm.

DOI: https://doi.org/10.7554/eLife.50231.030

The following video and figure supplements are available for figure 7:

**Figure supplement 1.** Correlative live-cell, SEM, and NanoSIMS imaging of mouse peritoneal macrophages, demonstrating that particles released onto the substrate during movement of filopodia and lamellipodia are enriched in accessible cholesterol but not sphingolipid-bound cholesterol.
DOI: https://doi.org/10.7554/eLife.50231.031

**Figure supplement 2.** Correlative SEM and NanoSIMS imaging, demonstrating that particles released onto the substrate by macrophages are enriched in accessible cholesterol but not sphingolipid-bound cholesterol.
DOI: https://doi.org/10.7554/eLife.50231.032

**Figure supplement 3.** The binding of fluorescently labeled OlyA to the plasma membrane overlaps the staining of the actin cytoskeleton with phalloidin.
DOI: https://doi.org/10.7554/eLife.50231.033

*Figure 7 continued on next page*

*Figure 7 continued*

**Figure supplement 4.** Correlative live-cell, SEM, and NanoSIMS imaging of mouse peritoneal macrophages treated with an FAK inhibitor, demonstrating that particles released onto the substrate during movement of filopodia and lamellipodia are enriched in accessible cholesterol but not sphingolipid-bound cholesterol.

DOI: https://doi.org/10.7554/eLife.50231.034

**Figure supplement 5.** Correlative live-cell, SEM, and NanoSIMS imaging of mouse peritoneal macrophages treated with an FAK inhibitor, demonstrating that particles released onto the substrate during movement of filopodia and lamellipodia are enriched in accessible cholesterol but not sphingolipid-bound cholesterol.

DOI: https://doi.org/10.7554/eLife.50231.035

**Figure supplement 6.** Correlative SEM and NanoSIMS imaging, demonstrating that particles released onto the substrate by macrophages treated with an FAK inhibitor are enriched in accessible cholesterol but not sphingolipid-bound cholesterol.

DOI: https://doi.org/10.7554/eLife.50231.036

**Figure 7—video 1.** Macrophage release particles during movement of filopodia and lamellipodia movement.

DOI: https://doi.org/10.7554/eLife.50231.037

**Figure 7—video 2.** Shows a macrophage treated with vehicle (DMSO) imaged by SEM and NanoSIMS in *Figure 7—figure supplement 2*.

DOI: https://doi.org/10.7554/eLife.50231.038

**Figure 7—video 3.** Shows a macrophage treated with an FAK inhibitor imaged by SEM and NanoSIMS in *Figure 7—figure supplement 4*.

DOI: https://doi.org/10.7554/eLife.50231.039

**Figure 7—video 4.** Shows a macrophage treated with an FAK inhibitor imaged by SEM and NanoSIMS in *Figure 7—figure supplement 5*.

DOI: https://doi.org/10.7554/eLife.50231.040

supplements *1–2*). NanoSIMS images revealed that [$^{15}$N]ALO-D4 bound strongly to the lawn of particles around macrophages; $^{15}$N enrichment was ~twofold higher in the lawn of particles than over the cell body (*Figure 8D*). $^{13}$C enrichment in the lawn of particles (from [$^{13}$C]OlyA binding) was very low (*Figures 7* and *8D*, *Figure 7—figure supplements 1–2*). We also observed differences in the patterns of [$^{15}$N]ALO-D4 and [$^{13}$C]OlyA binding to the cell body. In the case of [$^{15}$N]ALO-D4, we observed a fine linear band of $^{15}$N enrichment at the extreme outer edge of the cell (~40% greater $^{15}$N enrichment than over the cell body) (*Figures 7* and *8B–D*, *Figure 7—figure supplements 1–2*). In the case of [$^{13}$C]OlyA, a broad band of $^{13}$C enrichment was observed close to the outer edge of the cell (~60% higher $^{13}$C enrichment than over the main body of the cell) (*Figures 7* and *8B–D*, *Figure 7—figure supplements 1–2*). The broad band of $^{13}$C enrichment corresponded to the lamellipodia of macrophages, visible in the SEM images (*Figure 7*, *Figure 7—figure supplements 1–2*) as well as in the $^{12}$C$^-$, $^{12}$C$^{14}$N$^-$, and $^{32}$S$^-$ NanoSIMS images (*Figure 8A*). By confocal microscopy, the binding of OlyA at the outer edge of the cell overlapped with the binding of phalloidin to the actin cytoskeleton–rich lamellipodia (*Figure 7—figure supplement 3*). Interestingly, the fine linear band of $^{15}$N enrichment at the outer edge of the cell *extended beyond* the broad band of $^{13}$C enrichment, as shown by composite images formed from $^{15}$N/$^{14}$N and $^{13}$C/$^{12}$C ratio images (*Figure 8B–C*). We also performed correlative live-cell/SEM/NanoSIMS studies with macrophages that had been incubated with an FAK inhibitor (*Figure 7—figure supplements 4–6*, *Figure 7—videos 3–4*). The FAK inhibitor did not induce any changes to the patterns of [$^{15}$N]ALO-D4 and [$^{13}$C]OlyA binding to particles on the substrate or the macrophage cell body (*Figure 7—figure supplement 4–6*).

The fact that macrophages release cholesterol-rich particles inspired us to explore whether the efflux of [$^{3}$H]cholesterol from [$^{3}$H]cholesterol-loaded macrophages would be reduced when the release of particles was blocked with blebbistatin. To test this possibility, [$^{3}$H]cholesterol-loaded macrophages were plated on 6-well plates and incubated in serum-free medium (*i.e.*, no cholesterol acceptors) containing blebbistatin, an LXR agonist, or vehicle (DMSO) alone. After 24 hr, both the macrophages and macrophage-derived particles on the substrate were released with 5 mM EDTA. Both macrophage and particle fractions were prepared, and [$^{3}$H]cholesterol was quantified by scintillation counting. Compared with cells that had been incubated with DMSO alone, more [$^{3}$H]cholesterol was retained within the blebbistatin-treated macrophages (*Figure 8—figure supplement 1A*), and there was less [$^{3}$H]cholesterol in the particle fraction (*Figure 8—figure supplement 1B*). The LXR agonist had the opposite effect, reducing the amount of [$^{3}$H]cholesterol in macrophages and increasing the amount of [$^{3}$H]cholesterol in the particle fraction (*Figure 8—figure supplement 1A–*

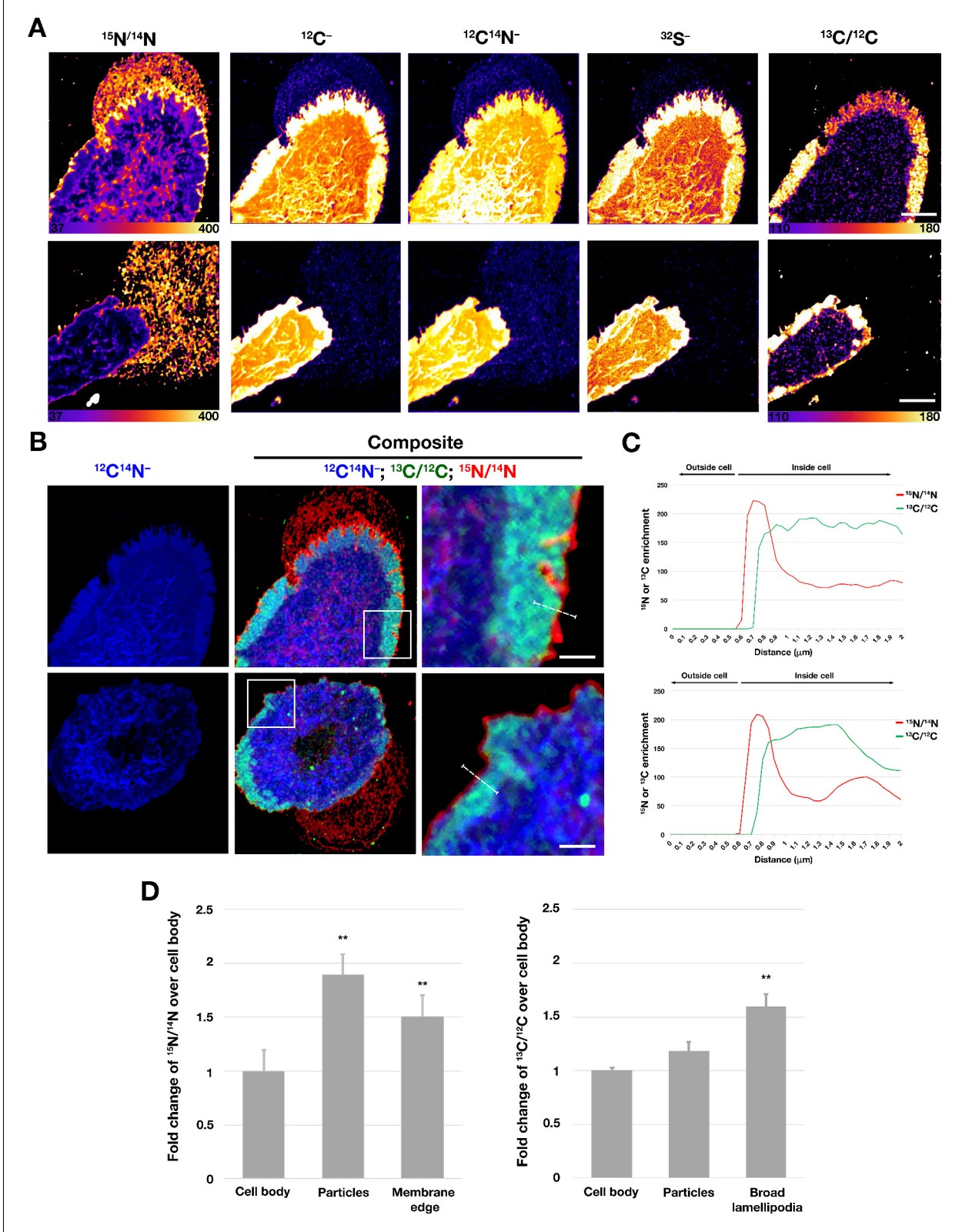

**Figure 8.** Distribution of distinct pools of cholesterol in the macrophage plasma membrane. Mouse peritoneal macrophages were plated onto poly-D-lysine–coated MatTek dishes and incubated in medium containing 10% FBS. On the next day, cells were incubated with [$^{15}$N]ALO-D4 and [$^{13}$C]OlyA (20 µg/ml each) and then imaged by NanoSIMS. (**A**) Distributions of [$^{15}$N]ALO-D4 and [$^{13}$C]OlyA binding to the plasma membrane. $^{13}$C/$^{12}$C NanoSIMS images revealed a broad band of $^{13}$C enrichment near the outer edge of the cell (corresponding to the flat lamellipodia of cells, apparent in the $^{12}$C⁻,

*Figure 8 continued on next page*

*Figure 8 continued*

$^{12}C^{14}N^-$, and $^{32}S^-$ NanoSIMS images). The $^{15}N/^{14}N$ images revealed a thin band of $^{15}N$ enrichment at the far outer edge of the cells and in the lawn of particles on the surrounding substrate. Additional NanoSIMS images of these cells are shown (with different scales) in *Figure 7* and *Figure 7—figure supplement 1*. (B) Composite $^{12}C^{14}N^-$ (*blue*), $^{13}C/^{12}C$ (*green*), and $^{15}N/^{14}N$ (*red*) NanoSIMS images, revealing $^{15}N$ enrichment at the outer edge of the plasma membrane and in the lawn of particles on the surrounding substrate. Of note, the $^{15}N$ enrichment at the outer edge of the cell extended beyond the thick band of $^{13}C$ enrichment. In this figure, the scales for the $^{13}C/^{12}C$ and $^{15}N/^{14}N$ NanoSIMS images were adjusted for optimal visualization of the lamellipodia of macrophages. Also, $^{12}C^-$ and $^{32}S^-$ images were included to visualize the lamellipodia. Scale bar, 5 μm. Additional NanoSIMS images of the cell in the top row of this panel are shown in *Figure 8A* and *Figure 7—figure supplement 1*; additional NanoSIMS images of the cell in the bottom row of this panel is shown in *Figure 7—figure supplement 2*. Two independent experiments were performed; representative images are shown. (C) Line scans comparing the $^{15}N/^{14}N$ and $^{13}C/^{12}C$ isotope ratios over the outer edge of the plasma membrane (*white* line in the upper and lower right images of panel B). (D) Quantification, by NanoSIMS, of [$^{15}$N]ALO-D4 and [$^{13}$C]OlyA binding to macrophages and to the surrounding particles on the substrate. $^{15}N/^{14}N$ ratios were quantified for the cell body, macrophage-derived particles, and the thin line of $^{15}N$ enrichment at the edge of the cell. $^{13}C/^{12}C$ ratios were quantified for the cell body, macrophage particles, and the broad lamellipodia near the edge of the plasma membrane. For each category, twenty-five regions, 20 pixels in diameter areas were circled on the $^{12}C^{14}N^-$ images, and the $^{15}N/^{14}N$ and $^{13}C/^{12}C$ ratios in each circle were calculated. A minimum of six macrophage images were used. Graph shows the mean and standard deviation of the fold change of $^{15}N$ or $^{13}C$ enrichment, normalized to the macrophage cell body. **$p<0.001$.

DOI: https://doi.org/10.7554/eLife.50231.041

The following figure supplements are available for figure 8:

**Figure supplement 1.** Reduced efflux of [$^3$H]cholesterol from [$^3$H]cholesterol-loaded macrophages when the release of particles was blocked with blebbistatin.

DOI: https://doi.org/10.7554/eLife.50231.042

**Figure supplement 2.** Increased efflux of [$^3$H]cholesterol from [$^3$H]cholesterol-loaded macrophages in the presence of HDL.

DOI: https://doi.org/10.7554/eLife.50231.043

*B*). As an additional control, we tested whether adding HDL to the culture medium would increase [$^3$H]cholesterol efflux from [$^3$H]cholesterol-loaded macrophages. As expected, HDL markedly increased efflux of [$^3$H]cholesterol into the cell culture medium (*Figure 8—figure supplement 2*).

## Macrophages release cholesterol-rich particles onto a collagen matrix and onto dead cells

We also plated biotinylated mouse peritoneal macrophages onto glass-bottom MatTek dishes that were coated with polymerized collagen IV. The macrophages released biotinylated particles, detectable with streptavidin-conjugated 40-nm gold nanoparticles, onto the collagen fibers (*Figure 9A*, *Figure 9—figure supplement 1*). To test whether the particles were enriched in accessible cholesterol, we plated biotinylated macrophages onto glass coverslips coated with PFA-fixed Alexa Fluor 647–labeled collagen IV and then incubated the cells with Alexa Fluor 488–labeled ALO-D4 and Alexa Fluor 568–labeled streptavidin. By STED microscopy, binding of ALO-D4 to the lawn of particles on the collagen matrix surrounding macrophages colocalized with streptavidin binding (*Figure 9B*). The amount of collagen beneath the lawn of particles was depleted, likely reflecting digestion of the collagen IV by the filopodia/lamellipodia of the macrophage.

Macrophages also release particles onto dead cells. By live-cell microscopy, we observed a *live* macrophage 'carrying' a *dead* macrophage, allowing us to observe the projection and retraction of lamellipodia/filopodia over the surface of the dead macrophage (*Figure 10—video 1*). We then imaged the cells by SEM. By SEM, we observed release of particles onto both the substrate surrounding the live macrophage and to the surface of the dead macrophage (*Figure 10—figure supplement 1*). In follow-up studies, we plated biotinylated macrophages onto a confluent monolayer of dead endothelial cells (fixed with 0.1% glutaraldehyde). As expected, we observed binding of streptavidin-conjugated gold nanoparticles to the cell body and filopodia of macrophages; however, we also observed binding of streptavidin-conjugated gold nanoparticles to the surface of adjacent endothelial cells (*Figure 10*). Although we could easily find gold nanoparticles on the endothelial cells in these studies, we were unable to identify with confidence macrophage-derived membrane particles on the surface of endothelial cells (because of the complicated topography of the plasma membrane of glutaraldehyde-fixed endothelial cells). Nevertheless, the distribution of gold

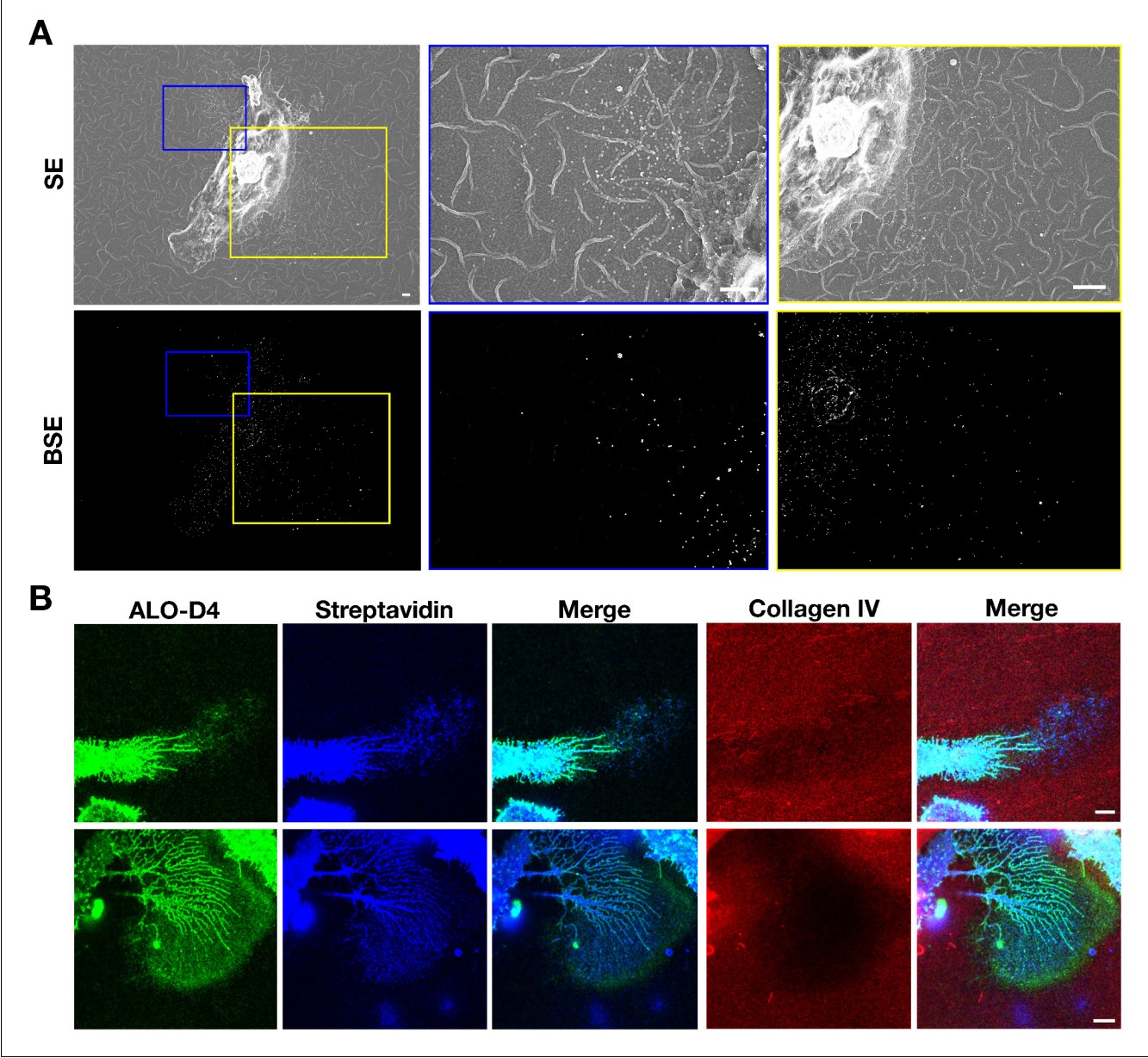

**Figure 9.** Mouse peritoneal macrophages release accessible cholesterol–enriched particles onto a polymerized collagen IV matrix. After biotinylating the cell-surface proteins of macrophages with Sulfo-NHS-SS-biotin, the cells were plated onto glass-bottom Petri dishes that had been coated with polymerized Alexa Fluor 647–labeled collagen IV and then fixed with 0.1% glutaraldehyde (for SEM) or 1% paraformaldehyde (for fluorescence microscopy). (**A**) SEM images show binding of streptavidin-conjugated 40-nm gold nanoparticles to macrophages as well as particles on the collagen IV substrate. Secondary electron (SE) and backscattered electron (BSE) images show macrophage-derived particles on the collagen fibers and the binding of gold nanoparticles to the macrophage cell body, filopodia, and the plasma membrane–derived particles. BSE images were helpful to identify gold nanoparticles. Higher magnification images of the *yellow* and *blue* boxed regions are shown on the right. Scale bar, 2 μm. (**B**) Fluorescent microscopy studies of mouse peritoneal macrophages that had been plated for 24 hr on a polymerized Alexa Fluor 647–labeled collagen IV substrate and then incubated with Alexa Fluor 488–labeled ALO-D4 and Alexa Fluor 568–labeled streptavidin. The cells were then fixed with 3% PFA and images recorded by STED microscopy. ALO-D4 (*green*) and streptavidin (*blue*) were visualized on a lawn of particles on the collagen IV matrix (*red*). Four independent experiments were performed; representative images are shown. Scale bar, 5 μm.

DOI: https://doi.org/10.7554/eLife.50231.044

The following figure supplement is available for figure 9:

**Figure supplement 1.** Mouse peritoneal macrophages release particles onto a polymerized collagen IV matrix.

DOI: https://doi.org/10.7554/eLife.50231.045

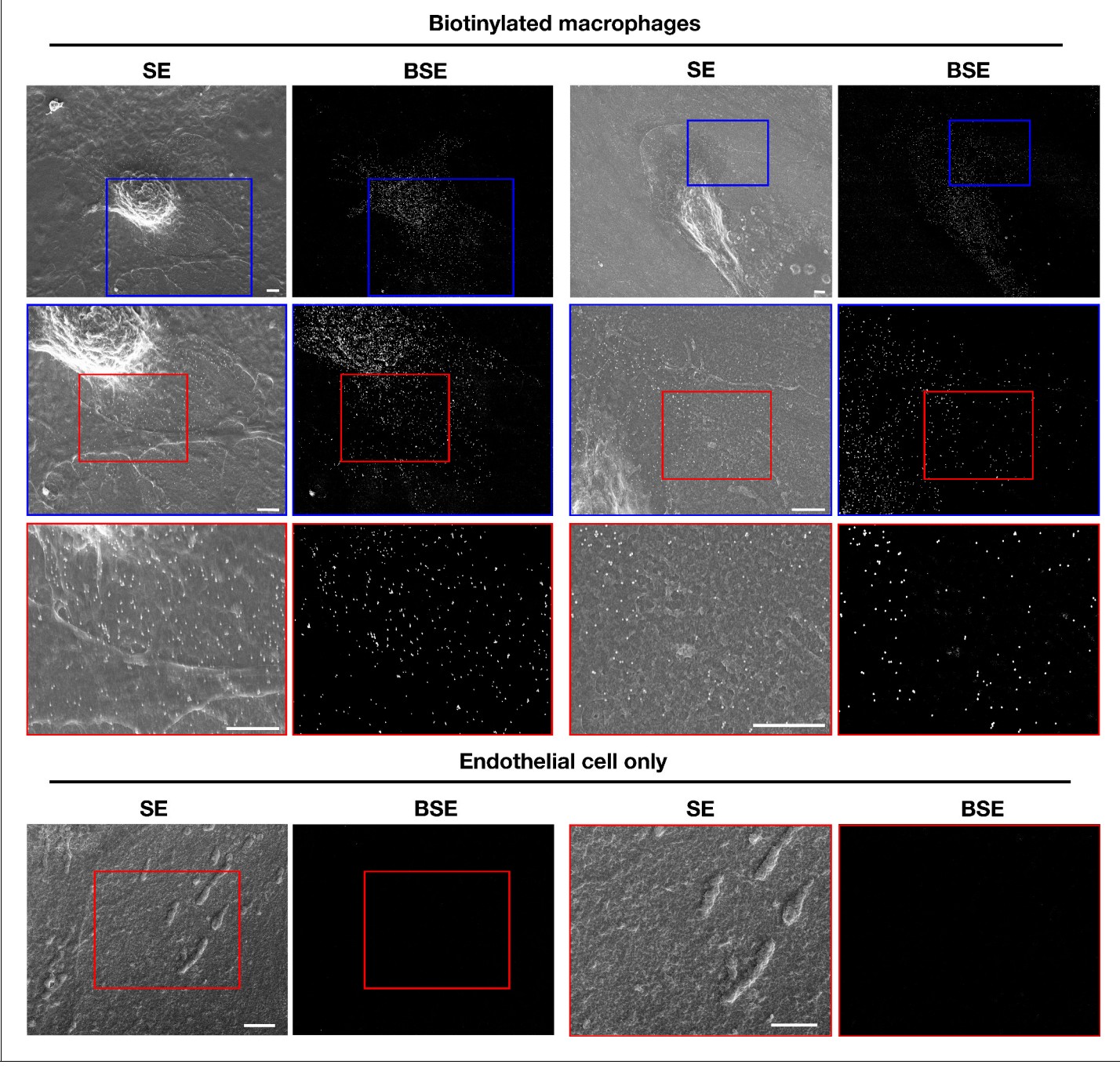

**Figure 10.** Biotinylated mouse peritoneal macrophages release plasma membrane–derived material onto the surface of dead endothelial cells. Mouse brain endothelial cells (bEnd.3) were plated onto glass- bottom Petri dishes and allowed to grow to confluency. After fixing the endothelial cells with 0.1% glutaraldehyde in PBS, they were washed extensively with PBS. Next, biotinylated macrophages (*i.e.*, macrophages in which the cell-surface proteins had been biotinylated with Sulfo-NHS-SS-biotin) were plated onto bEnd.3 cells. Secondary electron (SE) and backscattered electron (BSE) images were obtained with a scanning electron microscope. As expected, the SEM images revealed binding of the streptavidin- conjugated gold nanoparticles to both the cell body and filopodia of macrophages. In addition, gold nanoparticles were observed on the surface of adjacent endothelial cells. BSE images were helpful in identifying gold nanoparticles. Higher magnification images of the *blue* and *red* boxed regions in each image are shown below. As an experimental control, bEnd.3 cells without macrophages were fixed and incubated with streptavidin-conjugated gold nanoparticles. Higher magnification images of the *red* boxed regions are shown on the right. No binding of gold nanoparticles was detected. Three independent experiments were performed; representative images are shown. Scale bar, 2 μm.

DOI: https://doi.org/10.7554/eLife.50231.046

The following video and figure supplement are available for figure 10:

*Figure 10 continued on next page*

*Figure 10 continued*

**Figure supplement 1.** Correlative live-cell and SEM images, revealing the release of particles by a live macrophage onto the surface of a dead macrophage.

DOI: https://doi.org/10.7554/eLife.50231.047

**Figure 10— video 1.** Release of vesicular particles by a live RAW 264.7 macrophage onto the surrounding substrate and onto a dead RAW 264.7 macrophage.

DOI: https://doi.org/10.7554/eLife.50231.048

nanoparticles on the endothelial cells (adjacent to one pole of the macrophage) suggested that plasma membrane–derived biotinylated particles had been released onto the surface of endothelial cells.

## Discussion

In the current studies, we gleaned two insights into the genesis and composition of the ~30-nm vesicular particles that are released by cultured macrophages onto the surrounding substrate. First, we showed that particles are released from the macrophage plasma membrane during the projection and retraction of filopodia and lamellipodia. When the movement of filopodia/lamellipodia was blocked by actin depolymerization (latrunculin A) or by inhibiting myosin II (blebbistatin), particle release was abolished. Also, the deposition of particles onto the substrate occurred only in sites where we had observed (by live-cell microscopy) the projection and retraction of filopodia and lamellipodia. These findings implied that the particles were fragments of the plasma membrane that had been anchored to the substrate but subsequently torn away and left behind during the projection and retraction of filopodia/lamellipodia. Particles were also left behind as adherent macrophages progressively retracted during latrunculin A–induced depolymerization of the actin cytoskeleton. The fact that macrophage particles contained proteins associated with the plasma membrane, including adhesion complex proteins and cytoskeletal proteins, is consistent with the conclusion that the particles are plasma membrane fragments left behind during cellular locomotion. Second, we found that the plasma membrane–derived particles were enriched in 'accessible cholesterol' but not in sphingolipid-sequestered cholesterol. By super-resolution fluorescence microscopy and NanoSIMS imaging, we found robust binding of ALO-D4 (specific for the accessible pool of cholesterol) to the lawn of particles around macrophages. In contrast, there was minimal binding of OlyA, which binds to the sphingolipid-sequestered pool of cholesterol in the plasma membrane.

Our studies with biotinylated macrophages provided further evidence that the particles on the surrounding substrate are derived from the plasma membrane. After an overnight incubation, the lawn of particles around macrophages was readily detectable with fluorescently labeled streptavidin. Also, we observed binding of streptavidin-conjugated gold nanoparticles to the particles on the substrate around macrophages. Immunocytochemistry studies designed to detect specific plasma membrane proteins on the lawn of particles were inconclusive, very likely because of the low abundance of any given protein within a particle and because the binding affinities of most antibodies are far lower than the affinity of streptavidin for biotin.

We observed particularly large lawns of particles around macrophages that had been treated with an FAK inhibitor, which interferes with the disassembly of focal adhesions (the macromolecular complexes that tether cells to the underlying substrate). It makes sense that inhibiting focal adhesion disassembly would increase the likelihood of plasma membrane fragments being left behind during cellular locomotion. However, even though the lawn of particles surrounding FAK inhibitor–treated macrophages was consistently larger in multiple experiments, we are hesitant to draw ironclad conclusions, simply because we have not yet developed a realistic and practical strategy for accurately quantifying numbers of particles in the 'lawn' surrounding an individual macrophage (let alone the numbers of particles surrounding large numbers of macrophages on a cell culture plate).

In our studies, the particles on the substrate surrounding macrophages were enriched (relative to the plasma membrane over the cell body) in accessible cholesterol but depleted in sphingomyelin-sequestered cholesterol. This finding was evident by both fluorescence microscopy and NanoSIMS

imaging. Why the lawn of particles would be enriched in the accessible pool of cholesterol has not been established in an unequivocal fashion, but we would propose a potential explanation—one stimulated from the observations by Raghupathy and coworkers (*Raghupathy et al., 2015*) on the role of the cytoskeleton in organizing plasma membrane lipids. They proposed that the actin cytoskeleton is tethered, through adaptor proteins, to long acyl-chain–containing phosphatidylserines in the cytoplasmic leaflet of the plasma membrane, which interact with microdomains on the outer leaflet containing cholesterol and long acyl-chain–containing lipids (*e.g.*, sphingolipids). According to this model, sphingolipid-associated cholesterol on the outer leaflet of the plasma membrane associates, indirectly, with the actin cytoskeleton and thus would presumably remain tethered to the actin cytoskeleton during movement of filopodia and lamellipodia. According to this model, the *non-sequestered* 'accessible' pool of cholesterol, which is known to be mobile (*Infante and Radhakrishnan, 2017*), would be more likely than the sphingolipid-sequestered cholesterol to be left behind on the substrate during the movement of filopodia/lamellipodia. Our NanoSIMS images of [$^{15}$N] ALO-D4 and [$^{13}$C]OlyA binding are consistent with this model. We observed broad bands of $^{13}$C enrichment along the perimeter of macrophages, corresponding to the broad, flat, actin cytoskeleton–rich lamellipodia, but there was minimal $^{13}$C enrichment in the lawn of particles surrounding cells. In contrast, both the lawn of particles on the substrate as well as the extreme edge of the macrophage plasma membrane (from which particles are released) were enriched in $^{15}$N. The thin linear band of $^{15}$N enrichment at the far edge of the cell extends beyond the broad band of [$^{13}$C]OlyA binding. Thus, the plasma membrane at the far edge of the cell is enriched in accessible cholesterol. Presumably, the negligible amounts of $^{13}$C enrichment in the lawn of particles around macrophages is explained by the low amounts of $^{13}$C within the thin $^{15}$N-enriched region at the far edge of the plasma membrane.

We found that macrophages release, during movement of filopodia and lamellipodia, large numbers of ~30-nm 'accessible cholesterol'–rich particles onto the substrate. In earlier studies, other groups have uncovered evidence for the release of membranous fragments during cell migration. For example, *Schmidt et al. (1993)* described the ripping away of integrin-containing, cytoskeletal element–free membranous particles from rat fibroblasts during cell migration, but those particles (visible by light microscopy) were larger than we observed by SEM around cultured macrophages. *Regen and Horwitz (1992)* described release of trails of membranous particles from migrating chicken fibroblasts, but again, the particles were larger than we observed in macrophages. In these studies, the cholesterol content of particles was not investigated. *Vedhachalam et al. (2007)* provided suggestive electron microscopy evidence for the protrusion of cholesterol-rich exovesicular domains from the surface of J774 macrophages, but they did not describe the release of vesicular particles from those cells. They proposed that the exovesicular protrusions contained ABC transporters and were involved in the efflux of cholesterol onto apolipoprotein AI molecules in the biogenesis of HDL. Most recently, Li Yu's group described release of 'migrasomes' from filopodia in a variety of cell lines during cell migration (*Ma et al., 2015*; *Huang et al., 2019*). Migrasomes were described as cholesterol-rich micron-sized structures that, by electron microscopy, contained multiple 50–100-nm vesicular structures. Migrasome formation was blocked by cholesterol depletion. They did not describe 'lawns' of ~30-nm particles on the substrate surrounding their cultured cells. In our studies, we did not observe structures resembling migrasomes within the filopodia of cultured macrophages. Also, in an earlier study (*He et al., 2018*), we observed lawns of particles surrounding macrophages even after depleting the cells of cholesterol with an overnight incubation with HDL.

Our studies revealed that cultured macrophages release large numbers of cholesterol-rich particles onto a poly-D-lysine-coated substrate, but whether macrophages in living tissues release cholesterol-rich particles in the same manner has not yet been established. Detecting the release of plasma membrane–derived particles in vivo will be extremely challenging, in part because it will be very difficult to distinguish plasma membrane–derived particles from plasma lipoproteins and other membrane-bound particles (*e.g.*, exosomes). However, we did find that macrophages are capable of releasing plasma membrane–derived particles onto more physiologic substrates (*e.g.*, collagen matrix, dead cells), lending plausibility to the notion that particle release could occur in vivo. Particle release by macrophages in vivo could be relevant to cholesterol homeostasis. In terms of normal cholesterol physiology, it is conceivable that particle release could contribute to reverse cholesterol transport. In support of this idea, the release of [$^{3}$H]cholesterol by macrophages (incubated in the absence of cholesterol acceptors) was reduced by abolishing particle release with blebbistatin. An

earlier study (*He et al., 2018*) revealed that HDL is effective in depleting cholesterol from plasma membrane–derived particles. Thus, particle release could facilitate the return of accessible cholesterol into the circulation. In terms of disease, the release of cholesterol-rich membrane particles onto matrix proteins in the interstitial spaces of the arterial wall could, over a period of many years, contribute to the accumulation of extracellular cholesterol in atherosclerotic plaques.

# Materials and methods

**Key resources table**

| Reagent type (species) or resource | Designation | Source or reference | Identifiers | Additional information |
|---|---|---|---|---|
| Cell line (*M. musculus*) | RAW 264.7 | ATCC | Catalog No. TIB-71 RRID: CVCL_0493 | |
| Cell line (*M. musculus*) | bEnd.3 | ATCC | Catalog No. CRL-2299 RRID: CVCL_0170 | |
| Recombinant DNA reagent | ALO-D4 plasmid | PMID: 25809258 | | Dr. Arun Radhakrishnan (UT Southwestern) |
| Recombinant DNA reagent | OlyA plasmid | PMID: 30712872 | | Dr. Arun Radhakrishnan (UT Southwestern) |
| Chemical compound, drug | N-(3-Dimethylaminopropyl)-N'-ethylcarbodiimide hydrochloride (carbodiimide) | Millipore-Sigma | Catalog No. 03449 | |
| Chemical compound, drug | Glutaraldehyde 25% solution | Electron Microscopy Sciences | Catalog No. 16220 | |
| Chemical compound, drug | Osmium tetroxide 4% solution | Electron Microscopy Sciences | Catalog No. 18459 | |
| Chemical compound, drug | Paraformaldehyde 16% solution | Electron Microscopy Sciences | Catalog No. 15170 | |
| Chemical compound, drug | EMbed 812 | Electron Microscopy Sciences | Catalog No. 14120 | |
| Chemical compound, drug | Sodium cacodylate trihydrate | Electron Microscopy Sciences | Catalog No. 12300 | |
| Chemical compound, drug | Uranyl acetate | SPI-Chem | Catalog No. 02624AB | |
| Chemical compound, drug | Latrunculin A | Sigma | Catalog No. L5163 | |
| Chemical compound, drug | Blebbistatin | Abcam | Catalog No. ab120425 | |
| Chemical compound, drug | FAK inhibitor | Calbiochem | Catalog No. CAS 4506-66-5 | |
| Chemical compound, drug | LXR agonist | Sigma | Catalog No. G6295 | |
| Chemical compound, drug | HDL | Alfa Aesar | Catalog No. J64903 | |
| Chemical compound, drug | [$^3$H]cholesterol | PerkinElmer | Catalog No. NET139250UC | |
| Chemical compound, drug | Gold–streptavidin conjugation reagent | Abcam | Catalog No. ab186864 | |
| Chemical compound, drug | EZ-Link Sulfo-NHS-SS-biotin | ThermoFisher | Catalog No. 21331 | |
| Chemical compound, drug | Sphingomyelinase from *Staphylococcus aureus* | Sigma | Catalog No. S8633 | |
| Other | 35 mm glass-bottom gridded MatTek dish | MatTek | Catalog No. P35G-1.5–14-CGRD | |

## Mouse peritoneal macrophages

Wild-type C57BL/6 mice were injected intraperitoneally with 1 ml of 3% Difco Fluid Thioglycollate Medium (Becton Dickinson). Three days later, macrophages were harvested by peritoneal lavage with 10 ml of Dulbecco's Phosphate Buffered Saline (PBS). Cells were centrifuged at $400 \times g$ for 5 min at 4°C, incubated with red blood cell lysing buffer (Sigma), and washed two times with cold PBS. Macrophages were plated onto fetal bovine serum (FBS)-coated Petri dishes ($8 \times 10^6$ cells/dish) and incubated overnight in macrophage growth medium [Dulbecco's Minimal Eagle Medium (Gibco) supplemented with 1% sodium pyruvate and 1% glutamine] containing 10% FBS (Gemini GemCell). On the following day, macrophages were released from the dish by incubating the cells with PBS containing 5 mM ethylenediaminetetraacetic acid (EDTA) for 30 min at 4°C. For fluorescence microscopy, cells were plated onto glass coverslips in 24-well plates (75,000 cells/well). For scanning electron microscopy (SEM) and NanoSIMS studies, cells were plated onto 0.5 $cm^2$ silicon wafers in 24-well plates (75,000 cells/well). All substrates were sterilized and coated with 0.1 mg/ml of poly-D-lysine (Sigma).

## Preparation of ALO-D4 and OlyA

ALO-D4 and OlyA were produced in *Escherichia coli* using plasmids from Dr. Arun Radhakrishnan (UT Southwestern) (*Gay et al., 2015*; *Endapally et al., 2019*). $^{15}$N-labeled ALO-D4 and $^{13}$C-labeled OlyA were prepared as described previously (*He et al., 2018*; *He et al., 2017*). Briefly, ALO-D4 and OlyA were expressed in BL21(DE3) pLysS *E. coli* (Invitrogen) in 1 L of minimal medium containing 20.2 mM $^{15}$NH$_4$Cl (for $^{15}$N-labeled ALO-D4) or minimal medium containing 10% D-Glucose-$^{13}$C$_6$ (for $^{13}$C-labeled OlyA) at 18°C for 16 hr. Expression was induced by adding 1 mM isopropyl β-D-1-thiogalactopyranoside (IPTG) to the medium. Cells were pelleted, sonicated, and the lysate was centrifuged at 4°C. The supernatant fluid was mixed with 4 ml of HisPur Cobalt resin (50% bed volume; ThermoFisher). The mixture was loaded into a column and allowed to flow through by gravity. The column was washed, and [$^{15}$N]ALO-D4 and [$^{13}$C]OlyA were eluted with a buffer containing 300 mM imidazole. Eluates were concentrated to 1 ml with an Amicon 10 kDa cut-off filter (Millipore). The [$^{13}$C]OlyA was further purified by gel filtration on a Superdex 200 10/300 column (GE Healthcare). The purified preparations of [$^{15}$N]ALO-D4 and [$^{13}$C]OlyA were stored at 4°C.

## Correlative light microscopy, SEM, and NanoSIMS imaging

35-mm glass-bottom gridded Petri dishes (MatTek) were sputter-coated with ~4 nm of iridium with an ion-beam sputtering system (South Bay Technologies). Dishes were then washed three times for 5 min with 100% ethanol, air dried, and then coated with 0.1 mg/ml poly-D-lysine overnight at 4°C. On the next day, the dish was rinsed three times with sterile water; after drying, peritoneal macrophages were plated at 50,000 cells/dish. Live-cell videos were captured with a Zeiss LSM800 confocal microscope with a Plan Apochromat 20×/0.80 objective. The incubation chamber was maintained at 37°C and 5% CO$_2$ with TempModule S1 (Zeiss) and CO$_2$ Module S1 (Zeiss). Images were captured at 5 min intervals for 24 hr. Next, the cells were washed with PBS containing Ca$^{2+}$ and Mg$^{2+}$ (PBS/Ca/Mg) containing 0.2% bovine serum albumin (BSA) three times for 2 min and then incubated with [$^{15}$N]ALO-D4 (20 μg/ml in PBS + 0.2% BSA) for 2 hr at 4°C. In some experiments, [$^{13}$C]OlyA (20 μg/ml in PBS + 0.2% BSA) was also included in the medium. Next, cells were washed three times for 2 min with PBS/Ca/Mg + 0.2% BSA and then fixed with 4% paraformaldehyde and 2.5% glutaraldehyde (both from Electron Microscopy Sciences) in 0.1 M sodium cacodylate (pH 7.4; Sigma) for 1 hr on ice. Cells were washed with 0.1 M sodium cacodylate (pH 7.4) three times for 5 min, then fixed with 2% OsO$_4$ (Electron Microscopy Sciences) in 0.1 M sodium cacodylate on ice for 1 hr. Samples were rinsed three times for 5 min with distilled water, dehydrated with increasing amounts of ethanol (30, 50, 70, 85, 95, and 100%; 3 × 10 min), and air dried. The glass coverslip attached to the bottom of the Petri dish was removed with Coverglass Removal Fluid (MatTek) and placed onto a pin stub with Pelco colloidal silver (Ted Pella) and then coated with ~5 nm of iridium. Taking advantage of the gridded coverglass, the very same cells imaged by live-cell imaging were imaged with a Zeiss Supra 40VP scanning electron microscope with a 3-KeV incident beam. Next, the same cells were imaged with a NanoSIMS 50L instrument (CAMECA). Samples were scanned with a 16-KeV $^{133}$Cs$^+$ beam, and secondary electrons (SEs) and secondary ions ($^{12}$C$^-$, $^{13}$C$^-$, $^{12}$C$^{14}$N$^-$, $^{12}$C$^{15}$N$^-$, $^{32}$S$^-$) were collected. For NanoSIMS imaging on RAW 264.7 macrophages, a 50 × 50-μm region of the section

was pre-sputtered with a ~1-nA beam current (primary aperture D1 = 1) for 26 s (to remove the iridium coating). A 40 × 40-µm region was imaged with an ~1.3-pA beam current (primary aperture D1 = 2) and a dwell time of ~2.5 ms/pixel/frame for five frames. 512 × 512–pixel images were obtained. For NanoSIMS imaging on peritoneal macrophages, a 25 × 25-µm region was imaged with an ~3-pA beam current (primary aperture D1 = 2) and a dwell time of ~0.5 ms/pixel/frame for 18–20 frames. 512 × 512–pixel images were obtained. Images were prepared with the OpenMIMS plugin in ImageJ. [The last 15 frames of image sets were used to avoid any possibility of signals from surface contaminants and coating materials.] $^{15}$N/$^{14}$N and $^{13}$C/$^{12}$C ratios images were used to identify areas of stable isotope enrichment (from [$^{15}$N]ALO-D4 and [$^{13}$C]OlyA binding).

## Macrophage particle isolation

RAW 264.7 macrophages were plated onto ten T175 cell culture flasks (Corning) and grown overnight to ~80% confluency in macrophage growth medium containing 1% lipoprotein-deficient serum (LPDS; Alfa Aesar). On the next day, cells were washed two times with ice-cold PBS and then incubated for 30 min at 4°C in PBS containing 10 mM EZ-link Sulfo-NHS-SS-biotin (ThermoFisher). The biotinylation reaction was stopped with Quenching Buffer (ThermoFisher). Cells were then washed three times for 5 min with PBS, lifted by incubating the cells in PBS containing 5 mM EDTA, and centrifuged at 300 × g for 5 min. The supernatant fluid containing particles was centrifuged again at 3000 × g for 10 min to remove membrane debris and subsequently filtered through a 0.22-µm filter. The cell pellet was sonicated at low power (1.5) on ice for five cycles of 45 s on and 30 s off. The pellet was added to PBS containing 250 mM sucrose and centrifuged at 3000 × g for 10 min at 4°C. The supernatant fluid (containing the particles) and the plasma membrane extracts were incubated with 1 ml of NeutrAvidin beads (ThermoFisher) for 1 hr at 4°C. The beads containing particles or plasma membranes were then loaded onto 2 ml columns, and the samples were allowed to flow though. The columns were then washed three times (4 ml each) with PBS containing 0.2% Triton X-100. The particles and plasma membranes were eluted with 500 µl of PBS containing 50 mM dithiothreitol (DTT).

## Negative-stain transmission electron microscopy

PBS containing 50 mM DTT (5 µl) was pipetted directly onto a freshly glow-discharged copper grid that had been coated with formvar and carbon (Electron Microscopy Sciences); the PBS was then blotted off with Whatman #1 filter paper. Next, 5 µl of particle preparation, the plasma membrane preparation, or PBS alone was pipetted onto the grid and allowed to adsorb for 1 min before blotting off with filter paper. Next, 5 µl of 2% uranyl acetate was pipetted onto the grid and blotted off, followed by another 5 µl of 2% uranyl acetate, which was allowed to incubate for 1 min before being blotted off. Grids were imaged using an FEI Tecnai T12 set to 120 kV accelerating voltage equipped with a Gatan 2k × 2 k CCD detector.

## Drug treatment of cultured macrophages

Thioglycollate-elicited peritoneal macrophages were plated onto FBS-coated Petri dishes in macrophage growth medium containing 10% FBS overnight at 37°C. On the next day, macrophages were released by incubating the cells with PBS containing 5 mM EDTA. Macrophages were then incubated for 1 hr in suspension in macrophage growth medium containing 10% FBS and 5 µM latrunculin A (Sigma), 30 µM blebbistatin (Abcam), or 2 µM focal adhesion kinase (FAK) inhibitor (CAS 4506-66-5; Calbiochem). All drugs were diluted in DMSO. After 1 hr, macrophages were plated in drug-containing culture medium onto poly-D-lysine–coated substrates (silicon wafers or glass-bottom MatTek dishes for SEM and NanoSIMS imaging; glass coverslips for confocal fluorescence microscopy) and incubated for 24 hr. In some experiments, macrophages were plated onto poly-D-lysine–coated substrate and allowed to adhere for 1 hr in macrophage medium in the absence of drugs. After removing the medium, the cells were then incubated in drug-containing medium for 24 hr.

## Shotgun proteomics

Protein samples were resuspended in 8 M urea in 100 mM Tris, pH 8.5 and reduced, alkylated, and digested by sequential addition of lys-C and trypsin proteases as described (*Kaiser and Wohlschlegel, 2005*; *Wohlschlegel, 2009*). The sample was fractionated using reversed-phase

chromatography and then eluted into a Fusion Lumos tribrid mass spectrometer (ThermoFisher). MS/MS spectra were collected and analyzed with ProLuCID and DTASelect algorithms (*Xu et al., 2006*; *Tabb et al., 2002*). Database searches were performed against a mouse database. Protein and peptide identifications were filtered with a false-positive rate of <5%, as judged by a decoy database strategy. Normalized spectral abundance factor (NSAF) values were calculated as described (*Florens et al., 2006*). Analysis of other background contaminants was performed using CRAPome (*Mellacheruvu et al., 2013*). Gene-annotation enrichment analyses were performed with Enrichr (*Chen et al., 2013*; *Kuleshov et al., 2016*).

## Immunogold SEM studies

Mouse peritoneal macrophages were grown on FBS-coated Petri dishes overnight in macrophage growth medium containing 10% FBS. On the next day, cells were washed and then released by incubating the cells in PBS containing 5 mM EDTA for 30 min at 4°C. The cells were washed three times with PBS before incubating in PBS containing 0.25 mg/ml of Sulfo-NHS-SS-Biotin (ThermoFisher) for 20 min at 4°C (1.0 ml for each $1 \times 10^6$ cells). Cells were pelleted at $300 \times g$ for 5 min and washed three times with 10 ml of PBS before plating onto glass-bottom MatTek dishes at 50,000 cells/dish. The cells were incubated for 24 hr in macrophage growth media containing 10% FBS. On the next day, cells were washed three times with PBS/Ca/Mg and fixed with 4% PFA and 0.1% glutaraldehyde in PBS for 1 hr at 4°C. Cells were washed three times for 5 min with PBS/Ca/Mg containing 0.2% BSA, blocked with blocking buffer [PBS/Ca/Mg containing 5% donkey serum, 5% BSA, and 0.1% cold water fish skin gelatin (Electron Microscopy Sciences)] for 1 hr at room temperature, and then incubated with streptavidin-conjugated gold nanoparticles (1/50, diluted in blocking buffer; Abcam) for 2 hr at 4°C. The samples were then washed three times with blocking bluffer (5 min each); fixed with 1% glutaraldehyde in 0.1 M sodium cacodylate for 10 min on ice; washed five times with 0.1 M sodium cacodylate (2 min each); incubated with 2% osmium tetroxide in 0.1 M sodium cacodylate for 45 min on ice; washed three times (5 min each) with ice-cold water; and dehydrated with a series of graded concentrations of ethanol. Secondary electron and backscattered electron images were recorded with a Zeiss Supra 40VP scanning electron microscope with a 5-KeV incident beam with a backscatter detector.

## Immunocytochemistry of macrophage particles

Peritoneal macrophages were plated onto glass coverslips coated with 0.1 mg/ml poly-D-lysine. Cells were incubated for 24 hr in macrophage growth medium containing 10% FBS. In some experiments, the 10% FBS in the medium was replaced with 1% LPDS (Alfa Aesar) with or without 50 μg/ml of acetylated low-density lipoproteins (acLDL, Alfa Aesar). In other experiments, the cells (after 24 hr of growth in medium containing 10% FBS) were treated with sphingomyelinase from *Staphylococcus aureus* (100 milliunits/ml) for 30 min at 37°C. On the next day, cells were washed three times for 5 min with PBS/Ca/Mg containing 0.2% BSA and then incubated for 2 hr at 4°C with Alexa Fluor 488–labeled [$^{15}$N]ALO-D4 and Atto 647N–labeled [$^{13}$C]OlyA (both at 20 μg/ml, diluted in PBS/Ca/Mg + 0.2% BSA). In some experiments, cells were incubated with Alexa Fluor 488–labeled [$^{15}$N]ALO-D4 (20 μg/ml) and Atto 647N–labeled streptavidin (Sigma; 1/100). In other experiments, cells were incubated with Alexa Fluor 488–labeled [$^{15}$N]ALO-D4 (20 μg/ml) and an mCherry–lysenin fusion protein (10 μg/ml). After these incubations, the cells were washed three times for 2 min with PBS/Ca/Mg containing 0.2% BSA; fixed with 3% PFA; and mounted onto glass slides with Prolong Gold mounting media (ThermoFisher). Images were recorded with a Leica TCS SP8 STED 3X confocal microscope using a 100×/1.4 objective. Alexa Fluor 488 images were obtained with a 488 nm white light laser and a 592 nm depletion laser. Atto 647N images were obtained with a 647 nm white light laser and a 775 nm depletion laser. mCherry images were obtained with a 587 nm white light laser. Sequential scans were recorded at 2048 × 2048 pixels.

## Plating macrophages on collagen

Collagen IV from human placenta (Sigma) was labeled with an Alexa Fluor 647 fluorophore (ThermoFisher). The fluorescently labeled collagen IV (1 mg/ml) was added to glass-bottom MatTek dishes on ice. Dishes were incubated overnight at 37°C and 5% $CO_2$ to induce polymerization. On the next day, the excess collagen was removed, and the dishes were rinsed three times with PBS before fixing

the collagen with 0.1% glutaraldehyde in PBS (for SEM) or 1% PFA in PBS (for confocal microscopy) for 10 min at room temperature. Dishes were washed ten times for 6 min with PBS. Macrophage were then plated onto the collagen IV–coated dishes (50,000 cell/dish) in macrophage growth medium containing 10% FBS for 24 hr. On the next day, cells were either incubated with streptavidin-conjugated gold nanoparticles for SEM or Alexa Fluor 568–labeled streptavidin and Alexa Fluor 488–labeled [$^{15}$N]ALO-D4 for confocal fluorescence microscopy.

## Immunogold SEM of macrophages plated on fixed endothelial cells

Mouse brain microvascular endothelial cells (bEnd.3; ATCC #CRL-2299) were plated onto glass-bottom MatTek dishes in DMEM (ATCC) containing 10% FBS (ATCC), 1% sodium pyruvate, and 1% glutamine. The cells were allowed to grow to 100% confluency. Next, the cells were rinsed with PBS and fixed with 0.1% glutaraldehyde in PBS for 10 min at room temperature. Cells were then washed ten times with PBS (6 min each). Macrophages that had been biotinylated (50,000 cells/dish) were plated onto the bEnd.3 cells and incubated in macrophage growth medium containing 10% FBS (Gemini GemCell) for 24 hr. On the next day, the cells were processed for immunogold SEM, as described earlier.

## Efflux of [$^{3}$H]cholesterol from macrophages

Mouse peritoneal macrophage were plated on FBS-coated Petri dishes overnight in macrophage growth medium containing 10% FBS. On the next day, cells were washed three time with PBS and then incubated for 24 hr in macrophage growth medium containing 1% LPDS, [$^{3}$H]cholesterol (1 μCi/ml, PerkinElmer), and acetyl-LDL (20 μg/ml). On the following day, cells were washed three times (5 min each) with PBS and then released from the plate with PBS containing 5 mM EDTA. Cells were then plated onto 6-well plates (1 × 10$^{6}$ cells/well) and incubated for 24 hr in serum-free macrophage growth medium containing blebbistatin (30 μM), an LXR agonist (1 μM), or vehicle (DMSO) alone. As an additional control, the cells in one study were incubated with HDL (20 μg/ml, Alfa Aesar). On the next day, the medium was removed and saved for scintillation counting. The cells were then washed three times (5 min each) with PBS and released from the plate with PBS containing 5 mM EDTA. Cells were pelleted by centrifugation (300 × g for 5 min), washed, and resuspended in 500 μl of PBS. The supernatant fluid was centrifuged again (3000 × g for 10 min) to remove debris and then filtered through a 0.22-μm filter to obtain the particle preparation. Samples of the cells, the particles, and the medium (100 μl) were mixed with 4 ml of scintillation fluid (Optiphase Hisafe 3), and $^{3}$H dpm were recorded with a scintillation counter.

## Cell lines

RAW 264.7 and bEnd.3 cells were obtained from ATCC with proper 'certificate of analysis'. All cell lines were negative for mycoplasma contamination.

## Statistics

Statistical analyses of data were performed with GraphPad Prism 7.0 software. Quantitative data are reported as mean ± standard deviation. Differences were assessed with a Student's $t$-test with Welch's correction.

## Study approval

Animal housing and experimental protocols were approved by UCLA's Animal Research Committee. The animals were housed in an AAALAC (Association for Assessment and Accreditation of Laboratory Animal Care International)-approved facility and cared for according to guidelines established by UCLA's Animal Research Committee. The mice were fed a chow diet and housed in a barrier facility with a 12 hr light-dark cycle.

## Acknowledgements

This work was supported by grants from the NHLBI (HL090553, HL087228, HL125335) and a Transatlantic Network Grant from the Leducq Foundation (12CVD04). Xuchen Hu was supported by a Ruth L Kirschstein National Research Service Award (T32HL69766) and UCLA's Medical Scientist Training

Program. Haibo Jiang was supported by an Australian Research Council Discovery Early Career Researcher Award and a Cancer Council Western Australia Early Career Investigator Grant. STED microscopy was performed at the Advanced Light Microscopy Laboratory at the California NanoSystems Institute at UCLA (S10OD025017, CHE-0722519).

## Additional information

### Competing interests
Peter Tontonoz: Reviewing editor, *eLife*. The other authors declare that no competing interests exist.

### Funding

| Funder | Grant reference number | Author |
|---|---|---|
| National Heart, Lung, and Blood Institute | HL090553 | Stephen G Young |
| National Heart, Lung, and Blood Institute | HL087228 | Stephen G Young |
| National Heart, Lung, and Blood Institute | HL125335 | Stephen G Young |
| Fondation Leducq | 12CVD04 | Stephen G Young |
| Ruth L Kirschstein National Research Service Award | T32HL69766 | Xuchen Hu |
| Australian Research Council | Discovery Early Career Researcher | Haibo Jiang |
| Cancer Council Western Australia | Early Career Investigator Grant | Haibo Jiang |

The funders had no role in study design, data collection and interpretation, or the decision to submit the work for publication.

### Author contributions
Xuchen Hu, Conceptualization, Formal analysis, Validation, Investigation, Writing—original draft, Writing—review and editing; Thomas A Weston, Cuiwen He, Rachel S Jung, Patrick J Heizer, Brian D Young, Yiping Tu, Haibo Jiang, Investigation, Writing—review and editing; Peter Tontonoz, James A Wohlschlegel, Supervision, Writing—review and editing; Stephen G Young, Conceptualization, Resources, Formal analysis, Supervision, Writing—original draft, Writing—review and editing; Loren G Fong, Supervision, Writing—original draft, Writing—review and editing

### Author ORCIDs
Xuchen Hu https://orcid.org/0000-0002-0944-624X
Peter Tontonoz http://orcid.org/0000-0003-1259-0477
Haibo Jiang https://orcid.org/0000-0002-2384-4826
Stephen G Young https://orcid.org/0000-0001-7270-3176
Loren G Fong https://orcid.org/0000-0002-4465-5290

### Ethics
Animal experimentation: Animal housing and experimental protocols were approved by UCLA's Animal Research Committee (ARC; 2004-125-51). The animals were housed in an AAALAC (Association for Assessment and Accreditation of Laboratory Animal Care International)-approved facility and cared for according to guidelines established by UCLA's Animal Research Committee.

### Decision letter and Author response
Decision letter https://doi.org/10.7554/eLife.50231.051

Author response https://doi.org/10.7554/eLife.50231.052

## Additional files

### Supplementary files

• Transparent reporting form
DOI: https://doi.org/10.7554/eLife.50231.049

### Data availability

All data generated or analysed during this study are included in the manuscript and supporting files.

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
