## [Decision Letter]

Thank you for submitting your article "Release of cholesterol-rich particles from the macrophage plasma membrane during movement of filopodia and lamellipodia" for consideration by *eLife*. Your article has been reviewed by three peer reviewers, including Fredric B Kraemer as the Reviewing Editor and Reviewer #1, and the evaluation has been overseen by Didier Stainier as the Senior Editor. The following individual involved in review of your submission has agreed to reveal their identity: Paul Dawson (Reviewer #1).

The reviewers have discussed the reviews with one another and the Reviewing Editor has drafted this decision to help you prepare a revised submission.

Summary:

Questions regarding the molecular mechanisms underlying cholesterol excretion remain an important missing link in our understanding. Young and coworkers recently employed state-of-the-art approaches to provide compelling evidence for a pathway by which membrane particles containing significant amounts of accessible cholesterol are released from the filopodia of macrophages. This pathway operates in parallel with the previously characterized ABC transporter-mediated movement of cellular cholesterol to HDL (although the activity of ABC transporters may contribute to the cholesterol content of those particles).

The present study uses live cell microscopy, proteomics, and NanoSIMS imaging to identify the source of those accessible cholesterol-enriched particles. The authors make the surprising and novel discovery that the particles represent membrane fragments (enriched in focal adhesion-related proteins) deposited by filopodia and lamellipodia in the process of cellular locomotion.

All the reviewers agreed that this is a very clear and well written manuscript that identifies the mechanism for the release of small cholesterol-rich particles from macrophage plasma membranes and that the methodologies utilized are state-of-the-art and provide convincing data supporting the authors conclusions.

Although the studies raise a number of questions, such as the generalizability of the process to other cells, the fate of the particle-associated cholesterol in vivo, why the enrichment in "accessible" cholesterol, the reviewers do not feel that additional experiments are currently required.

Essential revisions:

It was noted that a substantial body of literature has been overlooked by the authors in which other investigators have previously reported loss of fragments of cell membrane and attachment proteins during cell movement or excretion of microparticles (see Regen and Horwitz, 1992; Schmidt et al., 1993; Nandi et al., 2009, J Lipid Res 50:456, to cite a few). While the more extensive characterization of these membrane fragments by more modern techniques in the current manuscript provides additional information, it should be presented within the context these previous observations.

---

## [Author Response]

Essential revisions:It was noted that a substantial body of literature has been overlooked by the authors in which other investigators have previously reported loss of fragments of cell membrane and attachment proteins during cell movement or excretion of microparticles (see Regen and Horwitz, 1992; Schmidt et al., 1993; Nandi et al., 2009, J Lipid Res 50:456, to cite a few). While the more extensive characterization of these membrane fragments by more modern techniques in the current manuscript provides additional information, it should be presented within the context these previous observations.

We thank the reviewers for this comment. In the Discussion section of the revised manuscript, we have added a paragraph in which we have done our best to place our observations in the context of earlier observations on the release of membranous particles from migrating cells. As the reviewer noted, there are important similarities between our observations and the observations in earlier papers, but there are also important differences. Also, our studies were the first to use NanoSIMS and electron microscopy to study the mechanism for the release of large numbers of ~30-nm cholesterol-rich particles from macrophage filopodia and lamellipodia.